# Evolutionary and functional impact of common polymorphic inversions in the human genome

Carla Giner-Delgado [1,2,13], Sergi Villatoro [1,13], Jon Lerga-Jaso[1,13], Magdalena Gayà-Vidal [1,3], Meritxell Oliva[1], David Castellano [1], Lorena Pantano [1], Bárbara D. Bitarello [4], David Izquierdo [1], Isaac Noguera[1], Iñigo Olalde[5], Alejandra Delprat[1], Antoine Blancher[6,7], Carles Lalueza-Fox [5], Tõnu Esko[8], Paul F. O'Reilly [9], Aida M. Andrés[4,10], Luca Ferretti [11], Marta Puig [1] & Mario Cáceres [1,12]

Inversions are one type of structural variants linked to phenotypic differences and adaptation in multiple organisms. However, there is still very little information about polymorphic inversions in the human genome due to the difficulty of their detection. Here, we develop a new high-throughput genotyping method based on probe hybridization and amplification, and we perform a complete study of 45 common human inversions of 0.1–415 kb. Most inversions promoted by homologous recombination occur recurrently in humans and great apes and they are not tagged by SNPs. Furthermore, there is an enrichment of inversions showing signatures of positive or balancing selection, diverse functional effects, such as gene disruption and gene-expression changes, or association with phenotypic traits. Therefore, our results indicate that the genome is more dynamic than previously thought and that human inversions have important functional and evolutionary consequences, making possible to determine for the first time their contribution to complex traits.

[1] Institut de Biotecnologia i de Biomedicina, Universitat Autònoma de Barcelona, Bellaterra, Barcelona 08193, Spain. [2] Departament de Genètica i de Microbiologia, Universitat Autònoma de Barcelona, Bellaterra, Barcelona 08193, Spain. [3] CIBIO/InBIO Research Center in Biodiversity and Genetic Resources, Universidade do Porto, Vairão, Distrito do Porto 4485-661, Portugal. [4] Department of Evolutionary Genetics, Max Planck Institute for Evolutionary Anthropology, Leipzig, Saxony 04103, Germany. [5] Institute of Evolutionary Biology, CSIC-Universitat Pompeu Fabra, Barcelona 08003, Spain. [6] Laboratoire d'immunologie, CHU de Toulouse, IFB Hôpital Purpan, Toulouse 31059, France. [7] Centre de Physiopathologie Toulouse-Purpan (CPTP), Université de Toulouse, Centre National de la Recherche Scientifique (CNRS), Institut National de la Santé et de la Recherche Médicale (Inserm), Université Paul Sabatier (UPS), Toulouse 31024, France. [8] Estonian Genome Center, Institute of Genomics, University of Tartu, Tartu 51010, Estonia. [9] Social, Genetic, and Developmental Psychiatry Centre, Institute of Psychiatry, Psychology and Neuroscience, King's College London, London SE5 8AF, UK. [10] UCL Genetics Institute, Department of Genetics, Evolution and Environment, University College London, London WC1E 6BT, UK. [11] Big Data Institute, Li Ka Shing Centre for Health Information and Discovery, University of Oxford, Oxford OX3 7LF, UK. [12] ICREA, Barcelona 08010, Spain. [13] These authors contributed equally: Carla Giner-Delgado, Sergi Villatoro, Jon Lerga-Jaso. Correspondence and requests for materials should be addressed to M.C. (email: mcaceres@icrea.cat)

In the last decade a great effort has been devoted to characterizing all the variation in the human genome[1–5], which opens the door to determining the genetic basis of phenotypic traits and disease susceptibility. Nevertheless, despite the initial expectations, a significant fraction of the genetic risk for common and complex diseases is still unexplained[6,7]. Furthermore, not all variants have been studied at the same level of detail. In particular, inversions are a type of structural variant that changes the orientation of a genomic segment, usually without gain or loss of DNA, and they often have highly-identical inverted repeats (IRs) at their breakpoints. These characteristics make inversion detection very challenging, even with next-generation sequencing methods, and they have been largely overlooked in humans[8,9].

Genome-wide inversion discovery has been typically based on genome sequence comparison[10,11] or paired-end mapping (PEM)[4,12], although recent studies have exploited newer techniques that could be especially useful for inversion detection, such as long-read sequencing[13–15], Strand-seq[16], BioNano optical maps[17], or a combination of them[18]. In most cases around 100–200 inversions have been predicted, with a maximum of 786 predictions in the 1000 Genomes Project (1000GP)[4,19]. However, these methods are not suitable for high-throughput genotyping, and with few exceptions[4,16,20], just a reduced number of individuals (1–15) have been analyzed. Moreover, the presence of repetitive sequences at the breakpoints influences the inversions that can be detected by each technique and results in high error rates for inversion validation compared to other variants[4,18,20,21].

Apart from the intensely-studied 17q21.31 and 8p23.1 inversions[22,23], genotyping efforts have been restricted to a small number of inversions and samples. For example, four other large inversions have been genotyped by FISH[24] and five smaller inversions by PCR[25] in 27 and 42 individuals of four populations, respectively. In addition, PCR and inverse PCR (iPCR) have been used for targeted studies of 34 inversions in 70–90 Europeans[21,26] and a more worldwide characterization of three inversions[25,27,28]. Also, although inversion genotypes might be predicted based on SNP data, these methods can only detect inversions above a certain size or associated with specific SNP combinations and the error rate can be high[21,23,25,29]. Therefore, it is not yet clear how many polymorphic inversions really exist in humans and very little is known about their global frequency and distribution[19].

Actually, inversions have been a model in evolutionary biology for almost 90 years[30,31] and there are numerous examples of their phenotypic consequences and adaptive significance in diverse organisms, from plants to birds[32]. One of their main effects is related to recombination, since single crossovers within the inverted region in heterozygotes generate unbalanced gametes and, at the same time, the resulting inhibition of recombination could protect favorable allele combinations[30,31]. In addition, inversion breakpoints can directly alter the expression patterns of adjacent genes[9,33].

From the little information available, it is clear that inversions can have important consequences in humans[9]. Inversions are associated with haemophilia A[34], increased risk of neurodegenerative diseases[35–37], autoimmune diseases[23,29] or mental disorders[38]. They could also predispose to other genomic rearrangements with negative phenotypic consequences in the offspring[9]. Moreover, there is evidence that the 17q21.31 inversion increases the fertility of the carriers and has been positively selected in Europeans[22]. Finally, inversions have been shown to affect gene expression[23,28,29,39]. However, most of these effects are associated with just the two best-known inversions. Attempts to associate inversions with gene-expression and phenotypic variation in large datasets have been limited to those with simple breakpoints, and only a couple of additional candidates have been identified so far[4,40,41]. Thus, specific genotyping studies of a diverse range of inversions in multiple individuals are necessary to determine their functional and evolutionary impact.

Here, we have developed a new high-throughput genotyping method and we have characterized in detail 45 common polymorphic inversions. By combining accurate inversion genotypes in 551 individuals of different populations and the available genomic information, we show that a large fraction of inversions are not linked to other variants and have occurred recurrently. In addition, several of them have signatures of selection and/or functional effects, emphasizing the role of inversions in the human genome.

## Results

**High-throughput genotyping of inversions**. We focused on a representative set of 45 paracentric inversions from the InvFEST database[19], which comprised most of those experimentally validated by PCR-based techniques when the project started[19] and corresponds approximately to half of the estimated number of real variants with >5% frequency in human populations[4,14] (Supplementary Fig. 1, Supplementary Data 1). These inversions were originally detected in the comparison of the hg18 and HuRef genome assemblies[10] or a fosmid PEM survey in nine individuals[12], and between 1 and 36 of them have been identified in different recent studies (Supplementary Data 1). The main limitation for inversion genotyping was due to breakpoint IRs, that had to be of less than 25–30 kb and with target sites for certain restriction enzymes at both sides but not within the IRs[26], which excluded previously-known large inversions mediated by big repeat blocks[19]. Overall, the studied inversions are located throughout the genome (37 in the autosomes, 7 in chr. X and 1 in chr. Y), with sizes ranging from 83 bp to 415 kb. Also, 24 (53%) have been generated by non-allelic homologous recombination (NAHR) between >90% identical IRs (from 654 bp to 24.2 kb). The rest (47%) were probably generated by non-homologous mechanisms (NH), including 18 with small deletions or insertions in the derived allele that may have been created in a single complex FoSTeS/MMBIR event[4,21] (Supplementary Fig. 1, Supplementary Data 1).

Of those, 41 inversions were genotyped simultaneously using high-throughput assays derived from the multiplex ligation-dependent probe amplification (MLPA) technique[42]. For inversions with simple breakpoint sequences (17), we carried out directly custom MLPA assays with minor modifications. However, for inversions with repetitive sequences at the breakpoints (24), which are difficult to detect by most techniques, we developed a new method combining the principles of iPCR[26] and MLPA[42] named iMLPA. In both cases, two pairs of oligonucleotide probes were used to interrogate the two alternative orientations for each inversion, orientation 1 (O1) and orientation 2 (O2) (Fig. 1). Four additional inversions not initially included in the MLPA-like assays were tested independently by PCR or iPCR (Supplementary Data 1). The 45 inversions were genotyped in 551 individuals from seven populations studied in HapMap and 1000GP[1,3] with African (AFR) (YRI, LWK), European (EUR) (CEU, TSI), South-Asian (SAS) (GIH) or East-Asian (EAS) (CHB, JPT) ancestry, here referred as population groups (Supplementary Table 1).

MLPA and iMLPA inversion genotypes were carefully validated through several analyses and quality controls (Fig. 1): (1) comparison with 3377 available genotypes[19] (see Supplementary Data 1 for data source); (2) identification of potential iPCR or iMLPA problems caused by restriction site polymorphisms; and (3) association between inversions and other variants (see below). As part of the validation, we repeated by PCR or iPCR 2160 extra genotypes with discrepancies or possible errors plus

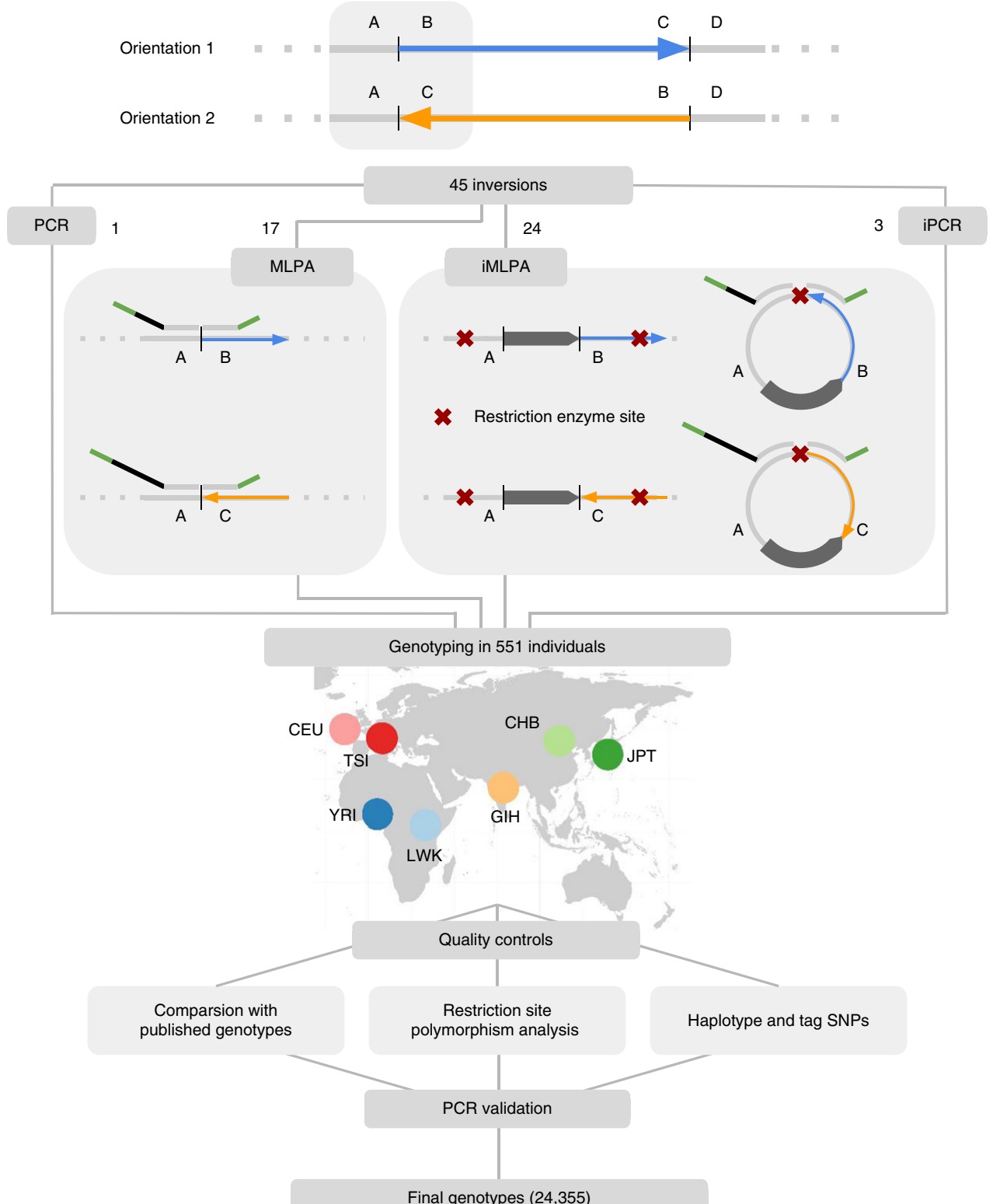

**Fig. 1** Schematic representation of inversion genotyping strategy. High-throughput genotyping of the 45 inversions in 551 individuals from different populations was done by MLPA (17), iMLPA (24), regular PCR (1) and iPCR (3). To avoid confusion about the ancestral status, the two inversion orientations have been named as 1 (AB and CD breakpoints) and 2 (AC and BD breakpoints), using the hg18 genome assembly as reference (Supplementary Data 1). In MLPA and iMLPA, two pairs of oligonucleotide probes (represented in top of the genome sequence) that are able to hybridize contiguously to the target region through a specific sequence complementary to the genome (light grey) were used to interrogate the two alternative orientations of each inversion. These probes, which include a stuffer sequence of variable length (black), are ligated together in a subsequent step and the resulting products are amplified for all the analyzed inversions at the same time with common primers (in green). IRs or other repetitive sequences at the breakpoints are represented as a dark pointed rectangle

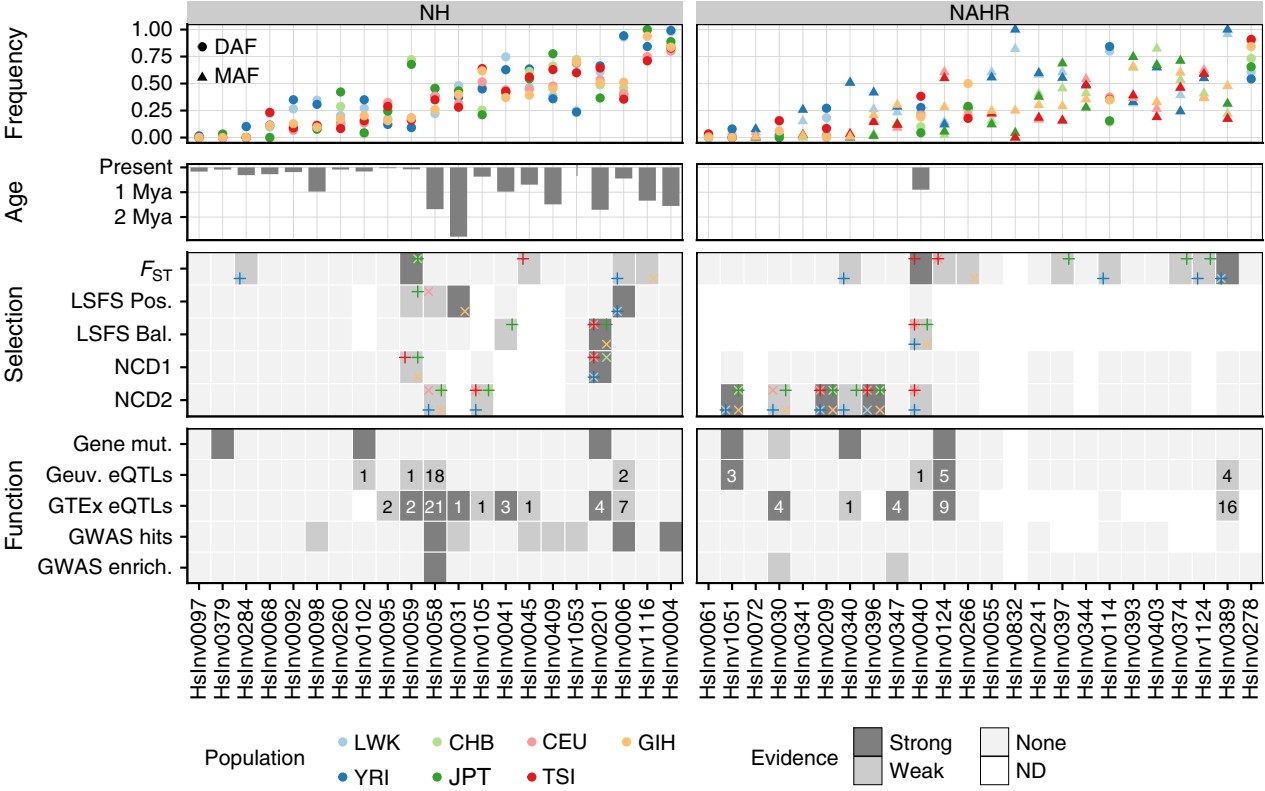

**Fig. 2** Evolutionary and functional information for human polymorphic inversions. Frequency: inversion frequencies in the 480 unrelated individuals from seven populations, showing either the derived allele frequency (DAF) if the ancestral orientation is known or the minor allele frequency (MAF) according to the global frequency of the inversion otherwise (which enables the MAF to be higher than 0.5 in specific populations). Age: average age for 22 inversions in which it can be calculated from the divergence between orientations using three different substitution rate estimates. Selection: summary of inversion selection signatures from $F_{ST}$, LSFS (positive or balancing selection) and NCD1 and NCD2 tests. Populations where the signal was detected are indicated by different colors in the corners of each cell, with alternating vertical and diagonal crosses to avoid visual overlap. Criteria for classification of strong and weak selection evidence are explained in Supplementary Data 7. Function: functional effects of inversions summarizing direct gene mutations, which include gene or transcript disruption (strong) or exchange of genic sequences (weak) (Supplementary Table 3), eQTLs in the GEUVADIS or GTEx datasets (showing the number of affected genes and labeled as strong if inversion is lead eQTL for at least one gene) (Supplementary Data 8 and Supplementary Data 9), and association with GWAS hits (strong if for one of the associations $P < 1 \times 10^{-6}$) (Supplementary Table 4) or GWAS signal enrichment (strong if enrichment empirical test $P < 0.01$ in both GWAS databases). Source data are provided as a Source Data file

others randomly selected, including the whole set of 551 individuals for the three inversions that had the highest error rate (HsInv0045, HsInv0055 and HsInv0340) (Supplementary Data 1). This showed that the new inversion genotyping technique is very robust, with missing data (0.7%) and genotype errors for MLPA (0.1%) and iMLPA (0.9%) accumulating mainly in specific problematic inversions or DNA samples (Supplementary Fig. 2A). When compared to the global inversion data from the 1000GP[4], just 14 of our inversions (31%) were detected, with nine of them having 2.5–71.5% incorrect genotypes and extremely low genotype agreement in the only two inversions mediated by large IRs in common (Supplementary Fig. 2B). Therefore, we have generated the largest and most accurate dataset of different types of inversions in humans (Supplementary Data 2).

As expected, the 45 inversions show correct genetic transmission in the 30 CEU and 30 YRI father-mother-child trios, and allele frequencies do not deviate from Hardy-Weinberg equilibrium in any population ($P > 0.01$). Minor allele frequencies (MAF) range globally from 0.5% to 49.7%, with 41 inversions spread through several population groups and only the four with the lowest frequencies being present in a single population group (HsInv0097, HsInv0284 and HsInv1051 in Africa; HsInv0379 in East Asia) (Fig. 2; Supplementary Table 1). On average, an

African and non-African individual carry the O2 allele, respectively, for 28 and 24 inversions.

**Nucleotide variation and haplotype distribution.** Thanks to the accurate genotypes, we were able to explore the linkage disequilibrium (LD) between inversions and neighboring variants (SNPs and small indels) from HapMap and 1000GP[1,3]. While most NH inversions (20/21) have variants in complete LD ($r^2 = 1$) either inside or up to 100 kb from the breakpoint, among the 24 NAHR inversions only HsInv0040 and HsInv1051 have at least one such variant (Fig. 3a; Supplementary Table 2). Maximum $r^2$ values between the remaining 22 NAHR inversions and 1000GP variants range from 0.14 to 0.91 (Fig. 3a).

Also, we checked the presence of shared SNPs (not including indels) in both orientations. Consistent with recombination inhibition in heterozygotes, most NH inversions do not have shared SNPs within the inverted region in accessible 1000GP positions or HapMap data, with the exception of a few individual SNPs that might be genotype errors or gene conversion events (Fig. 3a; Supplementary Table 2). Outside of the inversion, the average proportion of shared 1000GP SNPs increases progressively after the last fixed variant, up to ~20% (Fig. 3b). This allowed us to define an extended area on each

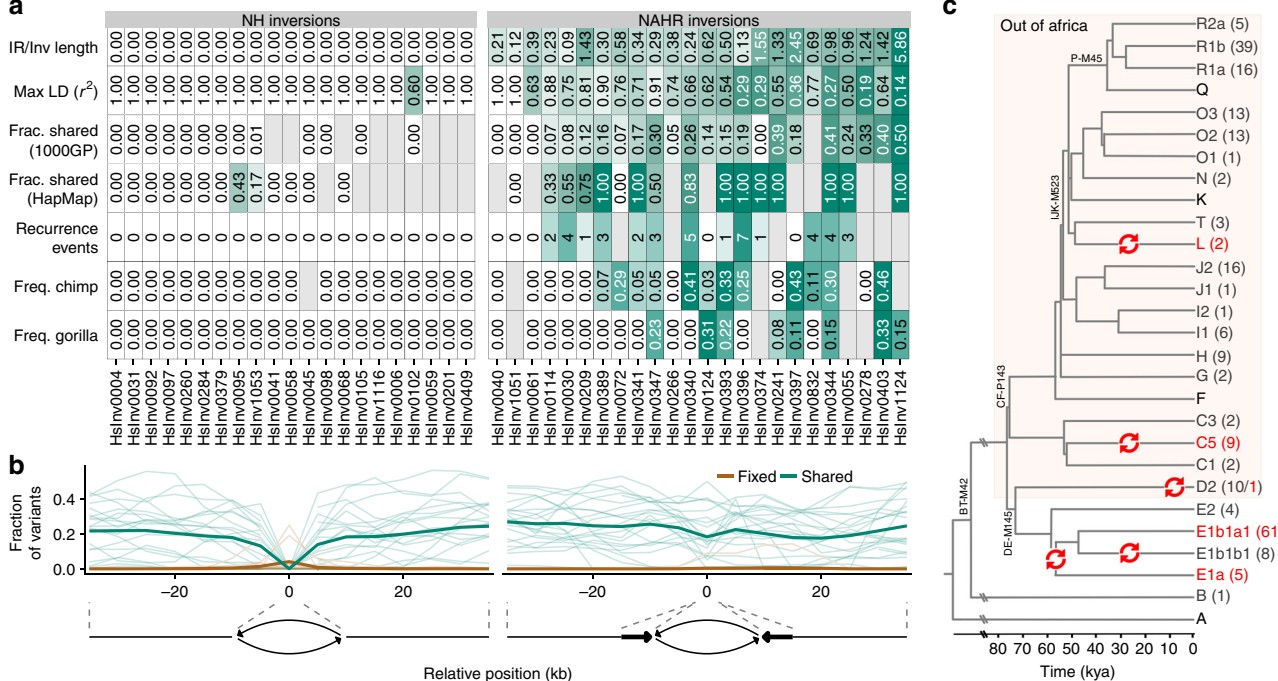

**Fig. 3** Evidence of unique or recurrent origin of NH and NAHR generated inversions. **a** Summary of the ratio between breakpoint IR and inversion length, maximum LD with neighboring variants in 1000GP and HapMap, fraction of shared SNPs inside the inverted region from the total number of SNPs analyzed in 1000GP or HapMap (which includes less SNPs and results in higher shared fractions), estimated number of recurrence events from haplotype analysis, and DAF or MAF for the inversions in non-human primates. Gray cells indicate values which could not be calculated. NAHR inversions show clear differences in LD with nearby variants, proportion of shared polymorphisms between orientations, recurrence events, frequency in ape species and other measures, with values that could be associated with higher levels of recurrence represented in stronger shades of green. **b** Distribution of fixed (brown) and shared (green) variants between the two orientations estimated from inversion genotypes and 1000GP data. The shared/fixed fraction with respect to the total number of polymorphic variants was computed for the whole inverted region plus 10-kb overlapping windows with a 5-kb step size in the flanking regions, and the horizontal axis represents the distance of the window central position to the inversion breakpoints (indicated by dashed lines). Thin lines represent individual inversions and thick ones represent the average for each inversion class. **c** Overview of the human chr. Y phylogenetic tree showing five different possible inversion events in the HsInv0832 region (red arrows). Divergence dates (bottom scale) and tree topology are based on Poznik et al[77]. For each chr. Y haplogroup, the number of males genotyped for HsInv0832 with each orientation is indicated in parenthesis (O1 in dark grey and O2 in red). Only the main branches and those including genotyped individuals are shown, with some characteristic mutations indicated in the tree. For haplogroup E, one of the two possible scenarios is represented, with the alternative being two inversion events in E1a and E1b1a1 haplogroups. Source data are provided as a Source Data file

side of the inversion of no or little recombination between orientations, which ranges from 0 to more than 20 kb (Supplementary Table 2). In contrast, 20 of the 24 NAHR inversions have a considerable number of shared SNPs scattered throughout the inverted and flanking regions (Fig. 3a, b; Supplementary Table 2).

Next, we visualized the haplotype diversity and distribution across orientations and populations using haplotype networks and a new representation integrating a hierarchical clustering of haplotypes and the differences between them (Supplementary Fig. 3). This analysis was focused on 1000GP data, although consistent results were obtained with HapMap SNPs. After taking into account possible phasing errors, two clearly differentiated patterns were observed again. In the 20 NH inversions with sequence variation information, the haplotypes of one of the orientations tend to cluster together, supporting a unique origin of the inversion (Supplementary Fig. 3). In NAHR inversions, this is true only for HsInv0040, HsInv0061 and HsInv1051, with the other 21 having O1 and O2 haplotypes mixed throughout the network and hierarchical cluster, including in many cases identical haplotypes with both orientations (Supplementary Fig. 3). Such pattern is consistent with a multiple origin of these inversions and explains the absence of fixed SNPs and the high number of shared SNPs between orientations.

Based on the results of the different analyses, we inferred the minimum number of recurrent inversion events and their distribution in human populations (Fig. 3a; Supplementary Data 3). However, this relies on having differentiated haplotype clusters in which the existence of the alternative orientation cannot be easily explained by other factors (such as gene conversion or genotype/phasing errors of a few variants). Another problem is that recombination generates mixed sequences between haplotype groups and makes it difficult to accurately quantify recurrence. Thus, a nice example is chr. Y inversion HsInv0832, in which there is no recombination. We used available haplogroup information of 232 males from the known chr. Y genealogical tree (Supplementary Data 4) to identify five independent inversion events in the last ~60,000 years (Fig. 3c). This results in an inversion rate of $5.35 \times 10^{-5}$ per generation (see Methods), which is ~1000 times higher than that of single bases. For the inversions in which it is possible to quantify recurrence, we estimated a total of 40 additional inversion and re-inversion events (ranging for each inversion from 0 to 7 with an average of 2.2) (Supplementary Data 3). Of those, 12 are distributed globally and could precede the out-of-Africa migration, 17 are restricted to African individuals, and 11 probably appeared more recently in non-African populations. The fact that many of the recurrence events are shared by several individuals indicates that they are not

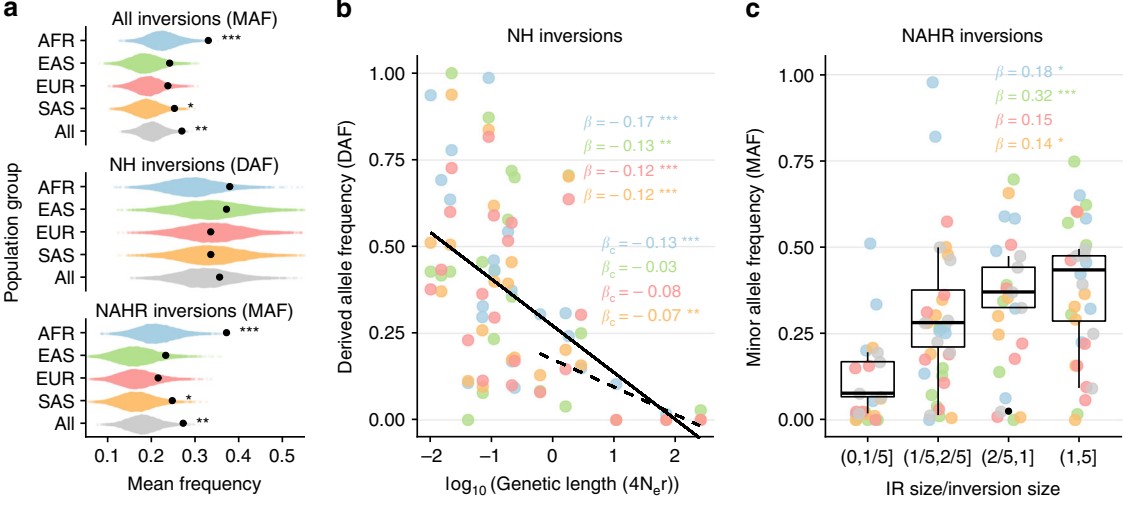

**Fig. 4** Determinants of inversion frequency in human populations. **a** Observed mean inversion frequency per population group and mechanism of generation (black dots) compared with that expected from the detection method simulations using a null distribution of 10,000 SNP samples. Graphs represent DAF only if the ancestral orientation is known for all the inversions included and MAF otherwise. **b** Logarithmic robust regression between DAF of NH inversions and genetic length (measured in $4N_er$ units) in the different population groups, showing a significant negative trend for all inversions (solid line; $\beta$ regression coefficient) and for inversions larger than 2 kb to correct in part the detection bias against low-frequency small inversions (dashed line; $\beta_c$ regression coefficient). **c** Boxplots showing a positive relationship between the frequency of the minor allele in all populations together and IR/inversion length ratio for NAHR inversions (centre, median; bounds of box, first and third quartiles; and whiskers, smallest/largest value within 1.5 interquartile range). $\beta$ indicates the robust regression coefficient in every population group. In **b** and **c** dots indicate the frequency of each inversion with population groups represented with the same colors as in **a**. Two-tailed $t$-test (**a**) and robust regression $t$-test (**b** and **c**) $P$ values: $*P < 0.05$; $**P < 0.01$; and $***P < 0.001$. Source data are provided as a Source Data file

artifacts from lymphoblastoid cell line (LCL) culture. In addition, as part of the validation of inversion genotypes, all the recurrence events were confirmed by checking at least one of the supporting individuals by PCR/iPCR. We have therefore extended considerably the previous recurrence analysis of some of the inversions in just the CEU population[21,26].

**Ancestral orientation and inversion age.** The published data on the ancestral orientation of 32 of the inversion regions[21,26,28] was complemented and expanded by experimentally testing 42 inversions for which the human or modified assays generated reliable results in a panel of 23 chimpanzees (40 inversions) and seven gorillas (41 inversions) (Supplementary Data 5). Inversion orientation was also assessed in available genome assemblies of both species plus orangutan and rhesus macaque (Supplementary Data 6). In total, we could infer the ancestral allele for 29 inversions, with 15 showing the ancestral and 14 the derived allele in the human reference genome. For the 21 NH inversions, orientation was consistent in all the non-human samples and genomes analyzed, and with existing deletions and insertions occurring in the derived allele (Supplementary Data 1 and Supplementary Data 6). In contrast, 14 of the 22 NAHR inversions experimentally genotyped were polymorphic in at least one of the apes (six in chimpanzees, two in gorillas and six in both species) and several had opposite orientation in different primate assemblies (Supplementary Data 6). This agrees with previous analyses in a much smaller set of samples[21,26], but we found five additional polymorphic inversions in each species. In fact, the lower number of polymorphic inversions in gorilla indicates that more inversion regions might be identified as polymorphic in non-human primates by analyzing more individuals. Given the species divergence times, the most likely scenario is that shared inversions have appeared independently in chimpanzee and gorilla lineages, providing additional support for inversion recurrence (Fig. 3a).

Moreover, we checked the presence of the breakpoint sequences of 19 NH inversions without IRs in available Neanderthal, Denisovan and two ancient modern human genomes (Supplementary Data 6). Five inversions (HsInv0004, HsInv0006, HsInv0201, HsInv0409, and HsInv1116) showed the derived orientation in the Neanderthal or Denisovan genomes (including two with the derived orientation in both). These inversions are distributed through African and non-African populations, which suggests that they are not the result of introgression and they appeared before the divergence of the most recent *Homo* groups, around 550,000–750,000 years ago (ya)[43].

Finally, we dated more precisely 22 unique inversions from the sequence divergence between orientations (Fig. 2; Supplementary Data 6). Six inversions were estimated to have appeared more than 1 Mya, including four with the derived orientation in Neanderthal or Denisovan, plus HsInv0031 and HsInv0058. For HsInv0006, the estimated age (407,795–495,470 ya) is slightly more recent than the Neanderthal-Denisova and modern human divergence, but it is very close to its lower bound. Ages of the rest of inversions are consistent with their geographical distribution, with inversions restricted to either African (HsInv0097 and HsInv0284) or non-African populations (HsInv0379) having a relatively recent origin. Only in two cosmopolitan inversions age estimates are lower than population split times (HsInv1053 with a negative age and HsInv0095 with 22,582–41,258 ya), probably due to an underestimation of the divergence between orientations caused by the limited sequence information available.

**Analysis of inversion frequencies.** To evaluate whether there are selective pressures acting globally on inversions, we compared the inversion frequency spectrum with that of SNPs sampled according to neutral expectations (see Methods). Although the ascertainment bias associated with inversion detection predicts an enrichment of high-frequency inversions (~0.20 expected MAF), the observed frequencies tend to be higher than expected

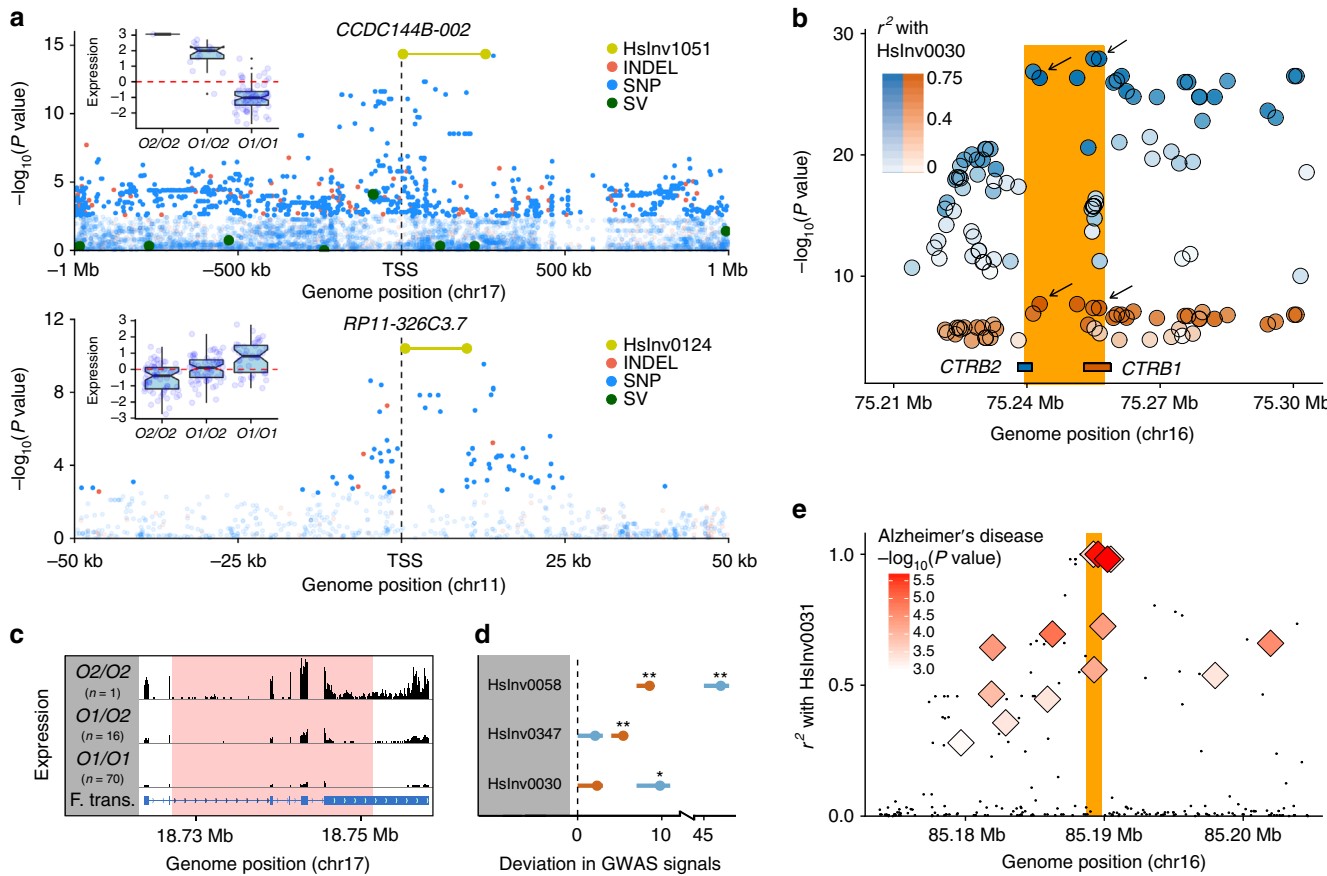

**Fig. 5** Examples of inversion functional effects. **a** Manhattan plots of logarithm-transformed linear regression t-test *P* values for *cis*-eQTL associations in LCLs of transcript *CCDC144B-002* and gene *RP11-326C3.7*, showing inversions HsInv1051 (top) and HsInv0124 (bottom) as lead variants, together with boxplots of rank normalized expression and inversion genotype (centre, median; bounds of box, first and third quartiles; whiskers, smallest/largest value within 1.5 interquartile range; and notches, 95% confidence interval of the median). **b** Manhattan plot of logarithm-transformed GTEx *P* values of pancreas eQTLs for *CTRB1* (red) and *CTRB2* (blue) mapping around the HsInv0030 region (orange bar). Top lead eQTLs for each gene were the variants in highest LD with the inversion (rs9928842, $r^2 = 0.75$; rs8057145, $r^2 = 0.73$; black arrows). **c** Schematic diagram of average RNA-Seq expression profile in HsInv1051 genotypes of GEUVADIS LCL reads mapped to the inverted allele, showing the generation of a fusion transcript (F. trans.) with new 3′ sequences that loses the last two thirds of the gene exons. Inversion breakpoint is indicated in pink. **d** GWAS Catalog (orange) and GWASdb (blue) signals enrichment within individual inversions and flanking regions (±20 kb). Dot represents the mean and error bars the 0–0.95 confidence interval of the difference between the observed number of GWAS signals and a null distribution from 1000 random genomic regions for each inversion. One-tailed empirical test *P* value: \*\**P* < 0.01; \**P* < 0.05. **e** LD ($r^2$) between HsInv0031 (orange bar) and 1000GP variants (black points) or Alzheimer's disease GWAS signals (diamonds)[78]. Source data are provided as a Source Data file

in all population groups (Fig. 4a). When inversions are separated by the generation mechanism, the increase in frequency is significant only in NAHR inversions (Fig. 4a). We also investigated how diverse genomic variables affect autosomal and chr. X inversion frequencies. The most significant predictor is inversion genetic length, which is negatively correlated with DAF of NH inversions and explains 23–55% of the frequency variance in the different population groups (Fig. 4b), followed by physical length (19–54% variance explained). For NAHR inversions, only 5–14% of MAF variance is accounted by their genetic length, whereas the ratio between IR and inversion length is positively correlated with inversion frequency in all population groups and explains 13–45% of MAF variance (Fig. 4c). This suggests that the higher frequency of NAHR inversions might be related to their repeated generation by recurrence.

**Selection on human inversions.** To investigate signals of natural selection acting on specific inversions, we first measured inversion frequency differences between populations using the fixation index

($F_{ST}$), which can identify positive selection leading to a rapid increase in allele frequency in some areas. The global $F_{ST}$ value was 0.11 for autosomal inversions, 0.21 for chr. X inversions, and 0.73 for the one in chr. Y, with the largest frequency variation between continents. Three inversions were within the top 1% of the $F_{ST}$ distributions derived from SNPs with the same frequency (Fig. 2; Supplementary Data 7): HsInv0040, with frequency differences between European populations, and HsInv0389 and HsInv0059, with high frequency in Africa or East Asia, respectively. Eleven more inversions fell within the top 5% of the empirical $F_{ST}$ distribution, and were considered to have weak evidence of selection (Fig. 2; Supplementary Data 7).

Second, we applied a novel test based on the frequency spectrum of linked sites (LSFS), which is well suited to detect deviations from neutrality in low-recombination regions, such as inversions. We used optimized tests to identify positive and long-term balancing selection maintaining a polymorphism in several populations, and significance was assessed empirically using the null LSFS distribution from autosomes. Only 18 unique autosomal inversions with nearby perfect tag variants

that could be reliably phased were analyzed, including most NH inversions and HsInv0040 (Fig. 2; Supplementary Data 7). The strongest signals (empirical test $P < 0.01$) were in HsInv0201 for balancing selection and HsInv0006 and HsInv0031 for positive selection. In addition, four other inversions showed weaker evidence of balancing or positive selection. Consistent with the $F_{ST}$ results, in HsInv0006 and HsInv0059 positive selection was detected in those populations with increased DAF (Fig. 2; Supplementary Data 7).

As independent confirmation of balancing selection signatures, we also used the recently developed non-central deviation statistics, NCD1 and NCD2, which detect site frequency spectrum shifts towards an equilibrium frequency and an excess of polymorphic sites[44]. However, the results of these tests summarize the data of all the SNPs in a region and are not necessarily linked to the inversion, as before. Focusing on signals detected in at least three populations, we found respectively four and six inversions with strong and weak signatures of balancing selection for NCD1 or NCD2 (Fig. 2; Supplementary Data 7). Many of these candidates could not be analyzed with the LSFS method because of the lack of tag SNPs or correspond to low-frequency inversions, such as HsInv1051 and HsInv0209, that are unlikely the targets of selection, but consistent results were found for HsInv0201.

**Effect of inversions on genes and gene expression.** As previously described, some of the analyzed inversions can have important effects on genes[4,9,21,25–28]. Although half of our inversions (21/ 45) are located in intergenic regions, three of them invert genes, eight are located within introns, seven might exchange gene sequences overlapping the IRs at the breakpoints, and six affect genes more directly through the inversion or deletion of an internal exon (HsInv0102, HsInv0201) and the disruption of the whole gene (HsInv0340, HsInv0379, HsInv1051) or an alternative transcript isoform (HsInv0124) (Fig. 2 and Supplementary Table 3).

We measured the effect of the 42 autosomal and chr. X inversions with MAF > 0.01 on expression of nearby genes (<1 Mb away) by linear regression between inversion genotypes and LCL transcriptome data from the GEUVADIS consortium[45]. To increase statistical power and reliability, the analysis was replicated in two datasets: (1) 173 CEU, TSI and YRI individuals with inversion genotypes; and (2) the complete GEUVADIS set of 445 European (358) and African (87) individuals in which the genotypes of 33 inversions could be imputed accurately (Supplementary Fig. 4A). Considering the largest sample size for each inversion, we uncovered eight inversions significantly associated with LCL expression of 27 genes and 44 transcripts (Supplementary Fig. 4B; Supplementary Data 8), with highly concordant results for those analyzed in both datasets (7/7 genes and 11/12 transcript effects were replicated) (Supplementary Fig. 4C). As negative control, no associations were observed by permuting inversion genotypes relative to expression levels (Supplementary Fig. 4D). Moreover, significant expression effects were robust when applying different analysis approaches (see Supplementary Fig. 4E-F and Supplementary Methods), and inversions acting as expression quantitative trait loci (eQTLs) located significantly closer to the transcription start site (TSS) of the differentially expressed genes (<100 kb) (Supplementary Fig. 4D).

Next, we examined inversion expression effects in other tissues through variants already reported as eQTLs in the GTEx project[46]. We found 62 genes with eQTLs in different tissues in high LD ($r^2 \geq 0.8$) with 11 of the 26 analyzed inversions, including seven not detected in LCL data (Supplementary Fig. 5;

Supplementary Data 9). By searching for eQTL signals in moderate LD ($r^2 \geq 0.6$) with some of the recurrent inversions, we found additional potential expression differences associated with HsInv0124 and in the genes affected by HsInv0030, which exchanges the first exon and promoter of chymotrypsinogen precursor genes *CTRB1* and *CTRB2* expressed only in pancreas[21,25], and HsInv0340, which disrupts the long non-coding gene *LINC00395* expressed in testis (Supplementary Fig. 5; Supplementary Data 9). In total, 17/27 of inversion-gene associations in LCLs were also identified in the smaller GTEx sample (Supplementary Fig. 5).

To assess if inversions were the main cause of the observed expression changes, we performed a joint eQTL analysis in LCLs including our inversions together with SNPs, indels and structural variants from the 1000GP[3,4]. Two inversions, HsInv0124 and HsInv1051, were the most likely causal variant for two genes and three transcripts in LCLs (Fig. 5a). Six other inversions show the highest LD ($r^2 \geq 0.9$) with variants reported as first or second lead eQTL in a given tissue by the GTEx project (Supplementary Fig. 6). Similarly, for recurrent inversions HsInv0124 and HsInv0030, eQTL significance in GTEx data increases with LD with the inversions, supporting their causal role (Fig. 5b). In general, some of the strongest effects are related to inversions affecting exonic sequences, although the consequences can be complex and need to be investigated in detail. For example, HsInv1051 breaks the *CCDC144B* gene and the apparent upregulation of specific isoforms (Fig. 5a) is actually due to the creation of a fusion transcript with new sequences at 3′ (Fig. 5c; Supplementary Fig. 7). HsInv0124 is the lead variant for the antisense RNAs *RP11-326C3.7* and *RP11-326C3.11*, which overlap respectively the *IFITM2* and *IFITM3* genes located at the breakpoints, and it has opposite effects in the two pairs of overlapping transcripts (Supplementary Data 8; Supplementary Data 9). Also, HsInv0102 removes the *RHOH* isoforms with the alternative non-coding exon that gets inverted, but its effect is masked by a more frequent lead eQTL (SNP rs7699141) acting in the same direction. On the other hand, HsInv0058 is associated with chr. 6 MHC haplotypes APD, COX, DBB, QBL and SSTO, which extend ~4 Mb and harbor important functional differences[47], suggesting that other variants in these haplotypes are responsible for the observed effects.

**Inversions and phenotypic traits.** The role of inversions in phenotypic variation was investigated using available genome-wide association studies (GWAS) data. We found a 1.26- and 1.95-fold increase in GWAS Catalog[48] and GWASdb[49] variants in the inversion and flanking regions (±20 kb). The top inversion driving this result was the MHC-inversion HsInv0058, but HsInv0030 and HsInv0347 showed similar enrichment of GWAS hits in both datasets (Fig. 5d; Supplementary Fig. 8A-B). GWAS signals close to the latter two inversions are consistent with their effect on genes, involving type 1 and 2 diabetes, pancreatic cancer, insulin secretion, and cholesterol and triglyceride levels for HsInv0030, and glaucoma and optic disc and nerve characteristics for HsInv0347, which is associated to the expression of *c14orf39* (*SIX6OS1*) and *SIX6*, related to eye development.

We also explored whether inversions were in strong LD ($r^2 \geq 0.8$) with known GWAS hits in the population where the association was reported. That is the case of HsInv0004, which is in complete LD in Europeans with a nearly genome-wide significant GWAS SNP related to asthma susceptibility in children and another one associated with body mass index in asthmatic children (Supplementary Table 4). Moreover, HsInv0006 is linked to schizophrenia in Ashkenazi Jews and glaucoma in Europeans (Supplementary Table 4). Remarkably,

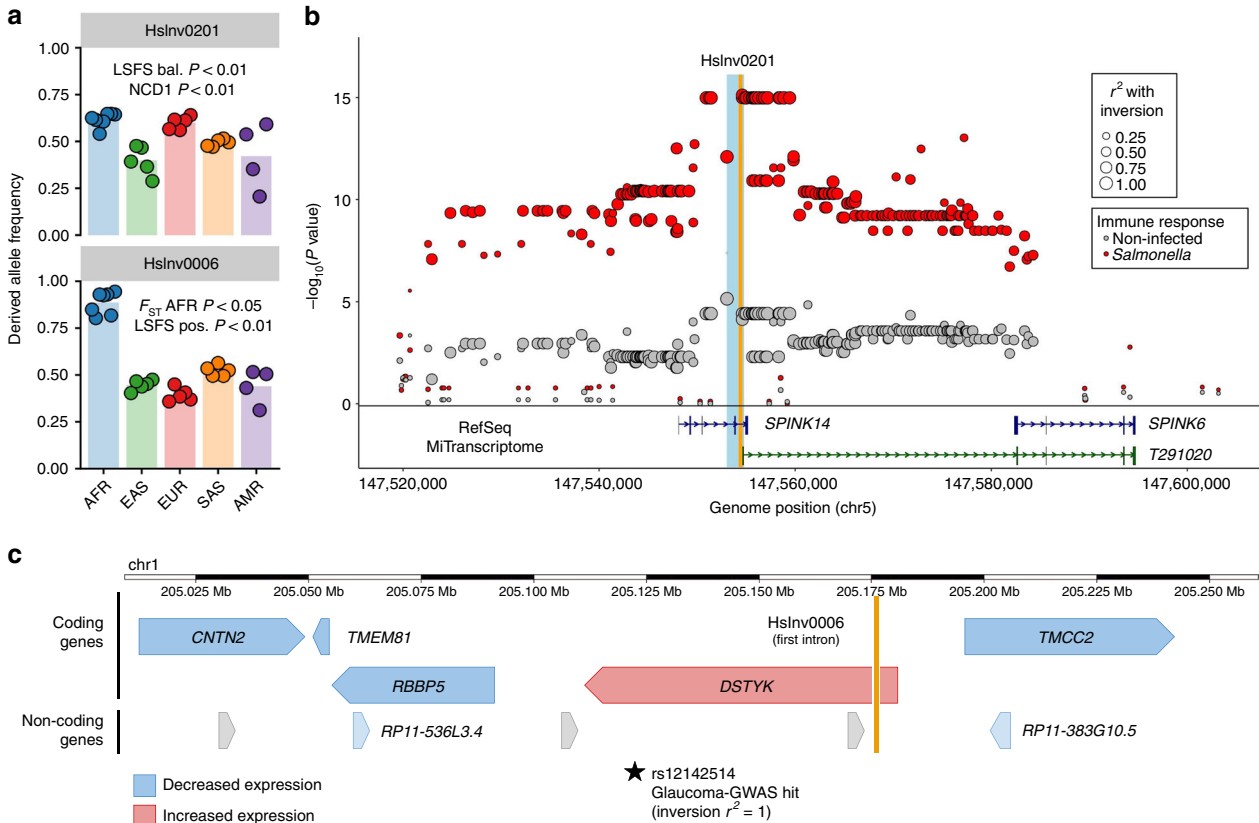

**Fig. 6** Integrative evolutionary and functional analysis of inversion candidates. **a** Frequency of HsInv0201 and HsInv0006 across worldwide human populations from 1000GP Ph3 (colored dots) and in each population group (colored bars) estimated from global inversion tag SNPs (rs200056603 and rs79619752, respectively), showing a summary of test results for balancing (bal.) and positive (pos.) selection. **b** Manhattan plot of logarithm-transformed linear regression t-test $P$ values for cis-eQTL associations with expression variation of gene *SPINK6* in infected and non-infected primary macrophages[52], showing SNPs in complete LD with HsInv0201 (orange bar) as lead eQTLs in *Salmonella* infection. Variants are represented as circles with varying size depending on the LD ($r^2$) with the inversion. The location of the genes in the region is shown below, including a new *SPINK6* isoform discovered by MiTranscriptome assembly[79], whose first exon is removed by a deletion associated to the inversion breakpoints (light blue bar). **c** Diagram of HsInv0006 (orange bar) genomic region showing the effect of the inverted allele on the expression of neighboring genes in different tissues according to the GTEx data and the inversion tag SNP in Europeans associated to increased risk of Glaucoma (Supplementary Table 4). Arrowheads indicate the direction of transcription of the genes. Source data are provided as a Source Data file

several of these inversions affect gene expression as well (Fig. 2). A good example is HsInv0031, which is associated with lower levels of *FAM92B* in cerebellum and is in almost perfect LD in Europeans ($r^2 = 0.98$) with SNP rs2937145 associated with Alzheimer's disease risk (Fig. 5e; Supplementary Table 4). Nevertheless, for many inversions the low LD with SNPs means that any effects would be missed in typical array-based GWAS (Supplementary Fig. 8C).

## Discussion

This work represents the most exhaustive and accurate study of human inversions so far, including a significant fraction of common inversions[19], and it is focused on inversions mediated by IRs, which generally escape detection. New genome-wide techniques are providing a more complete inversion catalogue[13–18], but they tend to be laborious and expensive, and population data in well-characterized individuals are still scarce. In addition, in many cases the high genotype error rate has precluded identifying tag variants that could be used to infer inversion effects[4,40]. Thus, the new genotyping method developed here and the reliable information from multiple individuals of different populations generated are crucial to fill the void in the knowledge of this type of variation.

Despite the effort to include as many inversions as possible, only those validated and with well-characterized breakpoints could be analyzed[19]. In addition, important limitations exist for the study of those mediated by IRs. Although it is not clear how many of them are real, in the InvFEST database there are ~100 inversion predictions with 1–25 kb inverted segmental duplications (SDs) at the breakpoints, which could be potentially interrogated using our method, and ~250 with larger SDs, which are currently out of reach of the methodology[19]. Therefore, further improvements and complementary strategies, such as the possibility of making directed cuts in specific positions or estimating the distance between regions separated by larger repeats, are necessary to expand the range of analyzed inversions. Nevertheless, the present work already offers a good picture of the contribution of common inversions to genetic diversity, adaptation, and phenotypic traits in humans.

In particular, we found clearly contrasting patterns for inversions generated by homologous and non-homologous mechanisms, supporting a high-degree of recurrence of all inversions mediated by highly-identical IRs except three (two of which have very low frequency). Recurrent inversions are characterized by low LD with other variants, a large amount of shared SNPs and shared haplotypes between *O1* and *O2* chromosomes (Fig. 3). Also, recurrence is not limited to humans but extends to other

great ape species. Similar results had been found in previous small-scale studies[21,24,26], but thanks to the analysis of many more individuals and populations, multiple controls (such as genotype confirmation by PCR of both breakpoints and different sources of nucleotide variation data) and a detailed analysis of haplotype relationships, we have obtained a better estimate of the independent inversion events. Specifically, we found nine more recurrence events in the five inversions originally predicted as recurrent in humans and that seven inversions considered to be unique or lacking information in CEU are recurrent as well[21,26]. This suggests that, like other repeats, IRs are rearrangement hotspots and the genome is more dynamic than previously thought. However, recurrence has been detected only through the sequences associated with the inversion, and we need more direct ways to quantify inversion generation rates precisely.

As for other types of recurrent changes[50,51], the lack of association between SNPs and many inversions and poor coverage by common arrays (Supplementary Fig. 8C) means that their phenotypic effects have been largely missed in GWAS. For example, half of the NAHR inversions cannot be accurately predicted with typical imputation algorithms (Supplementary Fig. 4A). We have found 23 and 22 inversions with different selection and functional signatures, respectively (Fig. 2). More importantly, although not all analyses could be applied to every inversion, there is a significant enrichment of inversions with both effects on genes or gene expression and selection signals directly linked to them (Fisher's exact test $P = 0.0320$) (see Supplementary Methods). This combination of the two independent types of evidence strongly indicates that inversions can have important consequences in humans.

One particularly interesting example is HsInv0201, an old inversion (>1.5 Mya) with intermediate frequency around the globe and clear signals of balancing selection (Fig. 6a), which deletes an exon of SPINK14 and is lead eQTL for two nearby genes (Supplementary Data 9). Moreover, the inversion haplotype is the main responsible for the lower SPINK6 expression during immune response to Salmonella infection[52,53] (Fig. 6b). In fact, HsInv0201 eliminates the promoter and first exon of a putative novel SPINK6 transcript (Fig. 6b) and it is in high LD ($r^2 = 0.971$ in EUR) with SNPs accounting for plasma levels of SPINK6 protein[54]. Together with the role of several of the affected genes in lung and extracellular mucosas, this suggests that the inversion could be related to immune response. In HsInv0006, its particular distribution pattern and the selection test results point to positive selection of the derived allele in Africa (Fig. 6a). Furthermore, the inversion is located within DSTYK first intron[21] and is associated with expression changes in the proximal genes, including DSTYK upregulation in different tissues (Fig. 6c; Supplementary Fig. 5). DSTYK deletion causes pigmentation problems and elevated cell death after ultraviolet irradiation[55]. Thus, positive selection on these traits could explain the inversion increase in Africa. Incidentally, the inverted orientation has been linked to higher risk of glaucoma in Europeans (Supplementary Table 4) and glaucoma is more common and severe in individuals from African ancestry[56]. Other interesting candidates include HsInv0031, HsInv0059, HsInv0124, HsInv0340, and HsInv0389 (Fig. 2).

Inversions differ from other genetic variants because of their expected negative consequences in fertility resulting from the generation of unbalanced gametes by recombination[30,31], which is exemplified by the reduction in frequency with genetic length (Fig. 4b) and the small number of inversions described compared to CNVs[19]. According to this, there could be a maximum length for an inversion to behave neutrally in terms of its fertility effects. Above that size, some type of compensatory selection, perhaps related to advantageous regulatory changes on nearby genes, would be necessary for the inversions to reach a certain

frequency. Therefore, this may explain the observed enrichment of inversions with functional and selection signals. Future in-depth studies of the identified candidates and other inversions will help uncover their real role in human evolution and the unexplained part of the genetic basis of complex traits.

## Methods

**Human and ape DNA samples.** We used 550 human samples included in the last phase of the HapMap Project and many of them in 1000GP phase 1 (Ph1) and 3 (Ph3)[1–3], which belong to seven populations of four main population groups (AFR, EUR, SAS, and EAS), plus individual NA15510 of unknown origin (see Supplementary Table 1 and Supplementary Data 2 for details). Most individuals were unrelated, but 70 are either children of mother-father-child trios (30 in YRI and 30 in CEU) or cryptic first and second-degree relatives (9 in LWK and 1 in GIH)[2,3]. Genomic DNAs of 70 CEU and 10 YRI samples and NA15510 were extracted from LCLs commercialized by the Coriell Cell Repository (Camden, NJ, USA), while the rest of DNAs were acquired from Coriell[21,26,28]. Chimpanzee and gorilla DNAs include six already used to genotype most of the inversions from frontal cortex tissue samples of the Banc de Teixits Animals de Catalunya (N457/03, Z01/03 and Z02/03) or LCLs from Barcelona Zoo individuals (PTR1211, PTR1213, and PTR1215)[21,26], plus 19 chimpanzee and five gorilla DNAs extracted from a previously-existing collection of primate LCLs of one of the authors (Supplementary Data 5). Ape samples comprise four mother-father-child trios and one father-son pair in chimpanzees and one father-son pair in gorillas. As for humans, cells were grown in 75-ml flasks to nearly confluency and high-molecular weight genomic DNA was obtained using a standard phenol-chloroform extraction[26]. All procedures that involved the use of human and non-human primate samples were approved by the Animal and Human Experimentation Ethics Committee (CEEAH) of the Universitat Autònoma de Barcelona.

**Experimental genotyping of inversions.** Initial genotypes for 41 inversions were obtained by newly developed assays based on the MLPA technique (Supplementary Data 1), which has been widely used for genome copy-number analysis and consists on the multiplex amplification of fragments of different sizes with common primers that are fluorescently labeled, and their detection by capillary electrophoresis[42]. Most NH inversions (17) were genotyped simultaneously from 100–150 ng of DNA with a slightly modified MLPA protocol using two pairs of probes that bind specifically at the breakpoint sequences of each orientation (AB and CD or AC and BD), with one of the probes that could be common to both pairs (Fig. 1). For 24 inversions with IRs or other repetitive sequences at the breakpoints, we developed a new iMLPA method that uses a combination of iPCR and MLPA. iMLPA requires some extra processing of the DNA with a restriction enzyme that cuts at each side of the breakpoint repetitive sequences, followed by self-circularization of all the digested DNA molecules together by ligation in diluted conditions with T4 DNA ligase and DNA purification. Then, MLPA was carried out as usual with a pair of probes that bind specifically at the self-ligation site of the circular molecules from each orientation (see Supplementary Methods for MLPA and iMLPA details). Supplementary Data 10 lists the sequences and concentrations of the probes used for MLPA (68) and iMLPA (87).

PCRs and iPCRs were carried out to genotype seven inversions in the 551 human samples (including four not analyzed in the MLPA/iMLPA assays) and to validate many of the MLPA/iMLPA genotypes (Fig. 1; Supplementary Data 1). Multiplex PCRs and iPCRs of each inversion were done with primers flanking either the breakpoint (PCR) or the self-ligation site of circularized molecules (iPCR) from the two orientations[21,26] (Supplementary Data 11). For the restriction site polymorphism analysis, we downloaded dbSNP (version 142) SNPs and indels around the inversion region (±50 kb) and determined all possible restriction site gains or losses affecting the iPCR/iMLPA experiments. To ensure that the genotypes were completely accurate, in most potential discrepancies both breakpoints of each orientation were tested.

For inversion genotyping in chimpanzees and gorillas, first we ran the same MLPA and iMLPA assays as described above, and those inversions that did not work were genotyped by PCR or iPCR. In some cases, new chimpanzee or gorilla specific primers and restriction enzymes for iPCR were used to overcome human assay problems[26] (Supplementary Data 5; Supplementary Data 11). However, this was not always possible due mainly to deletions or genome gaps, and a few inversions in one or both species could not be tested. All polymorphic inversions in chimpanzees or gorillas were validated by PCR or iPCR of at least one breakpoint to make sure that there were no errors in the iMLPA results.

**Analysis of nucleotide variants associated with inversions.** We measured pairwise LD ($r^2$) between inversions and overlapping and neighboring biallelic variants up to 200 kb at each side of the inversion from 1000GP Ph3 (including SNPs and indels for 434 unrelated individuals) and HapMap release 27 (including fewer SNPs but all the 480 unrelated individuals) using either plink v1.90[57] or Haploview v4.1[58]. Variants located within the breakpoint interval and associated deletions or IRs were excluded for this and subsequent analysis to avoid possible genotyping errors. Supplementary Data 12 lists the inversion tag variants with $r^2 \geq$

0.8 from 1000GP and HapMap data considering all the individuals together, as well as each population and population group. This analysis allowed us to detect a few inversion genotypes that did not match those expected from the tag variants and most of them were confirmed to be genotyping errors by independent PCR or iPCR validation (Supplementary Fig. 2). SNPs and indels around inversion regions were further classified directly from the genotype data according to its distribution across orientations in fixed ($r^2 = 1$), shared (unambiguously polymorphic in both orientations) and polymorphic in $O1$ or $O2$ chromosomes using in-house perl and bash scripts[21,26]. To minimize possible genotyping errors, only the most reliable variants according to the 1000GP strict accessibility mask were included. Non-recombining flanking regions were defined from the 1000GP data (which has more resolution than HapMap) as the longest sequence outside the breakpoints up to a maximum distance of 20 kb that: (1) does not contain reliable shared variants compatible with a crossing-over event between orientations; and (2) includes most of the fixed variants (Supplementary Table 2).

**Phasing and visualization of haplotypes.** Each orientation haplotypes were determined following two complementary strategies to minimize errors and obtain more robust results. First, after testing several commonly-used phasing programs, we selected PHASE 2.1[59] because it avoids switch errors in inversion heterozygotes by fixing the phase of the two orientation alleles added at the breakpoint positions[21,26]. Phasing was done independently for the YRI, LWK, EUR, SAS, and EAS populations or population groups, using the available trio information when possible and five iterations (−x5) for HapMap data and two iterations (−x2) with the hybrid model (−MQ) for 1000GP data. Only variants within the inverted region plus 20 kb of flanking sequence for 1000GP Ph1 data (which was the only one available at that time) or 200 kb for HapMap data were selected. Second, we took advantage of the 1000GP Ph3 phased haplotypes[3] to impute directly the inversion orientation based on the presence of perfect tag variants ($r^2 = 1$). For inversions without perfect tag variants, only homozygotes or hemizygotes for each orientation, in which the inversion status of the haplotypes can be assigned unambiguously, were analyzed.

The relationships between the different haplotypes of each dataset were visualized by combining several data sources and representation methods. Phased 1000GP Ph1 and HapMap haplotypes were used to build Median-Joining (MJ) networks[60] with the NETWORK v.4.6.1.3 software (www.fluxus-engineering.com). In addition, we devised our own representation of the haplotype relationships and the distribution of nucleotide changes along the sequence, named integrated haplotype plot (iHPlot), which combines a hierarchical clustering, distance matrix and visual alignment of the alleles in each polymorphic position, plus the haplotype orientation and the populations in which it is present (see Supplementary Methods for details).

Inversion origin was estimated from the information of the different phasing and haplotype visualization strategies, which overall showed consistent results (see Supplementary Methods). For inversions with perfect tag variants, the analysis was based mainly on 1000GP Ph3 haplotypes (including when possible the flanking non-recombining region), which allowed a better discrimination of haplotypes, filtering of accessible SNPs according to the pilot accessibility mask and had less phasing errors due to the use of sequences from more individuals[3]. For inversions without tag variants, we relied mainly in the phased 1000GP Ph1 haplotypes iHPlots, since all the genotyped individuals in common could be analyzed. Inversion recurrence events were conservatively estimated by identifying clusters of haplotypes with both orientations that differ significantly from all others with the same orientation as the potential recurrence event, after eliminating possible phasing errors in inversion heterozygotes (see Supplementary Methods). Around 1–25 individuals supporting each of the independent recurrence events were validated by PCR or iPCR (in this case testing mostly both breakpoints) to discard possible inversion genotyping errors (Supplementary Data 3). Also, we validated the genotype of many more individuals with unexpected orientation-haplotype combinations and of other inversions with a high proportion of shared SNPs for which recurrence events could not be clearly identified. HsInv0832 inversion rate was estimated from the publicly available information of 232 of the 282 genotyped males (Supplementary Data 4) by calculating the total number of mutations and generations along the genealogical tree of the analyzed Y chromosomes using the approach of Repping et al.[61] and Hallast et al.[62] (see Supplementary Methods).

**Ancestral orientation and inversion age estimate.** Besides the experimental analysis in chimpanzees and gorillas, the ancestral state of inversions was complemented by bioinformatic and manual inspection of four of the best non-human primate genome assemblies (see Supplementary Methods). Due to their fragmented status, the orientation of several available ancient hominin genomes (Supplementary Data 6) was determined by identifying the reads that span the $O1$ or $O2$ breakpoints of 19 NH inversions without IRs using a library with four 100-bp sequences centered at the two breakpoints of both orientations (Supplementary Data 13) and a slightly modified version of the BreakSeq pipeline[21,27,28].

Age of unique inversions was obtained with the usual divergence-based approach[63,64], using the pairwise differences between orientations from all SNPs (excluding indels) in the available 1000GP Ph3 haplotypes and the largest of the two average pairwise differences within $O1$ or $O2$ sequences. To have more information in short inversions, we considered the inverted region and any extra

non-recombining flanking region (up to 20 kb), excluding breakpoint intervals and associated IRs or indels to avoid sequence errors. Confidence intervals of age estimates were calculated by bootstrap sampling the same number of total individuals with replacement 1000 times and using both a constant substitution rate and two local substitution rates from the divergence with chimpanzee and gorilla (see Supplementary Methods).

**Inversion frequency analyses.** To control the effect of the study design ascertainment bias in the observed frequency of inversions, we simulated the detection and genotyping process in biallelic SNPs from 1000GP Ph3. The process was simulated in two steps: (1) selection of those SNPs for which the alternative allele is present in 1000GP individuals matching the demographic and gender composition of the detection panel (nine individuals for 38 autosomal or chr. X inversions detected from the fosmid PEM data[12] and one individual for six inversions detected exclusively from the genome assembly comparison[10]); and (2) for each of the PEM inversions, generate a random sample of 10,000 SNPs from the total pool of polymorphic SNPs in the PEM panel according to the simulated detection probability calculated from the SNP frequency and the inversion characteristics (see Supplementary Methods). Mean and median frequencies of inversions and SNPs in the 434 1000GP Ph3 individuals were compared by sampling 10,000 sets of SNPs without replacement, with one matched-SNP per inversion at a time, and empirical $P$-values were estimated as twice the fraction of samples with values more extreme or equal than the observed.

Different genomic variables were tested to determine their effect on inversion frequency: (1) physical length of the inverted region; (2) inversion genetic length; (3) number of genes within the inversion or breakpoint regions; (4) distance to the closest coding gene; (5) number of mammalian constrained sites inside the inversion;[65] (6) direct functional effect of inversions on genes (Supplementary Table 3); and (7) size of breakpoint IRs. Inversion genetic length was estimated as the cumulative $4N_er$ for all the genotyped chromosomes using the 1000GP Ph3 SNP data and the LDhat v2.2 rhomap function[66]. Due to high correlation between several variables, to keep only those with significant regression coefficients and reduce the effect of potential outliers in reduced samples, we built robust regression models of the autosomal and chr. X inversion frequency in each population group by stepwise forward selection of predictors with the lmrob function from robustbase R package[67].

**Inversion selection tests.** Frequency differences between populations were calculated with vcftools (v0.1.15) using the $F_{ST}$ statistic[68]. $F_{ST}$ values were compared with empirical null distributions in the same individuals obtained from biallelic SNPs accessible according to the strict mask, with a defined ancestral allele, and that have similar frequency and chromosome type (autosome or chr. X) as the inversion (see Supplementary Methods).

LSFS tests are a newly developed family of neutrality tests especially appropriate for inversions, which are a direct application of nearly optimal linear tests for neutrality[69]. The summary statistic was the frequency spectrum of variants closely linked to the inversion, including their linkage pattern (nested or disjoint) with the inverted allele[70], and we tested strong positive or balancing selection coefficients. LSFS was calculated from biallelic 1000GP Ph3 SNPs in the genotyped individuals, after removing those with a GERP score[65] higher than 2 and within 0.5 Mb of any of the inversions in our dataset. Only 18 autosomal inversions unambiguously phased into the 1000GP Ph3 haplotypes with perfect tag SNPs within 20 kb from the breakpoints were analyzed using 3 kb non-overlapping windows localized within the inverted or non-recombining flanking regions (skipping the breakpoints, IRs and indels to avoid genotype errors). Inversion windows were compared against the empirical LSFS computed around all autosomal SNPs and tests were conditioned on the inversion frequency in the different populations. Each population and window was tested separately and population $P$ values of the same windows were combined via Edgington's method[71], whereas the results across different windows of an inversion were combined using a conservative and an approximate approach (see Supplementary Methods).

NCD1 and NCD2 statistics[44] to test long-term balancing selection acting on autosomal and chr. X inversion regions were computed for three target frequencies (0.3, 0.4, and 0.5) in overlapping windows of 2 kb (with 1 kb step), defined with the same criteria as in the LSFS test, using 1000GP Ph3 SNPs (accessible according to the pilot mask) from all individuals of the seven studied populations. Only windows with a minimum of eight informative sites (either human polymorphisms or fixed differences with chimpanzee in NCD2 only) and at least 16.7% of positions covered by hg19-panTro4 alignments were considered. Finally, an empirical $P$-value was assigned for each inversion region and population by comparing the tests results with a null genome-wide distribution obtained by sampling regions of the same size as the inversion (see Supplementary Methods).

**Gene-expression analysis in LCLs.** We analyzed 42 inversions (excluding two with MAF < 1% and HsInv0832 in chr. Y) in 173 experimentally genotyped individuals (42 CEU, 84 TSI, 47 YRI) with GEUVADIS[45] and 1000GP Ph3[3] data. Besides, we imputed 33 inversions in the complete set of 445 GEUVADIS individuals in common with 1000GP Ph3 (89 CEU, 91 TSI, 86 GBR, 92 FIN, and 87 YRI) using the already identified perfect tag SNPs ($r^2 = 1$) (19 inversions) or

IMPUTE v2.3.2[72] (14 inversions with >90% average imputation accuracy) (Supplementary Fig. 4A). LCL raw RNA-Seq reads (ArrayExpress experiment E-GEUV-1) were aligned against the GRCh38.p10 human genome with STAR v2.4.2a[73] using GENCODE version 26 annotations[74]. Gene-expression levels were estimated as reads per kilobase per million mapped reads (RPKM) and transcript expression levels were quantified with RSEM v1.2.31[75]. *cis*-eQTL analysis was done through linear regressions implemented in QTLtools v1.1[76], considering 850 genes and 3318 transcripts expressed in at least 20% of the samples and with TSS within 1 Mb from inversions. First, we carried out a targeted study to test only the association with the genotypes of each inversion. Second, we performed a joint eQTL analysis with inversions and neighboring 1000GP variants to estimate their contribution to observed gene-expression changes and identify lead eQTLs. Expression values were adjusted by gender, the first three principal components from 1000GP data (corresponding to continent, population and population structure) and a set of expression principal components to capture technical confounding factors (for genes and transcripts, respectively, 10 and 20 in the experimental dataset and 35 and 50 in the imputed dataset) (see Supplementary Methods). Next, they were transformed to match normal distributions N(0,1) to avoid false positive associations due to outliers and significance was established at 5% false discovery rate (FDR).

**Inversions as eQTLs in other tissues and conditions**. Gene-expression effects in other tissues for 26 inversions were examined by looking whether their highest associated SNPs across all populations ($r^2 \geq 0.8$) have been identified as *cis*-eQTLs in different human tissues by the GTEx project (GTEx Analysis Release v7)[46]. Also, we extended the analysis to seven recurrent inversions with SNPs in moderate LD ($r^2 \geq 0.6$). To determine the potential causal variants, we checked if those SNPs in highest LD with the inversions were reported as being the first or second lead eQTL in a specific tissue. The same strategy was applied to link inversions to immune eQTLs associated to the transcriptional response to *Listeria* and *Salmonella* of primary macrophages from African-American ($n = 76$) and European-American ($n = 99$) individuals[52] and of macrophages differentiated from 123 induced pluripotent stem cell (iPSC) lines of European origin to *Salmonella* plus interferon γ stimulation[53].

**Association of inversions with GWAS SNPs**. Association of inversions with specific traits or diseases was based on the NHGRI Catalog of published GWAS (release 2017-07-17)[48] and the GWASdb (release 2015-08-19)[49] databases. To remove redundant entries, the strongest signal per locus (±100 kb genomic region) was selected. Only inversions in high LD ($r^2 \geq 0.8$) with a GWAS signal in the studied population or the closest one available were considered (Supplementary Table 4). To investigate if the number of GWAS signals in the inversion and flanking regions (±20 kb) was higher than expected by chance, we crossed GWAS Catalog and GWASdb signals with 1000GP variants and grouped together those in high LD ($r^2 \geq 0.8$) and associated to the same phenotype. Enrichment *P* values were calculated by comparison with a null distribution from 1000 random genomic regions as a background model for each inversion, controlling by inversion size and SNP frequencies (average SNP frequency ±0.01 and Chi-square test $P > 0.05$ for the number of SNPs with MAF <0.2 and ≥0.2 compared to the inversion region) and excluding gaps and chr. Y. To test the enrichment in specific inversions, we repeated the same analysis computing a one-tailed empirical *P*-value for each inversion.

**Statistical information**. Details of the statistical tests are described in the corresponding sections. In many cases *P*-values were derived from empirical genome-wide null distributions from at least 1000 random samples and two-tailed tests were always used, except when specifically mentioned.

**Reporting summary**. Further information on research design is available in the Nature Research Reporting Summary linked to this article.

## Data availability

All data described in this article are available in the Supplementary Information and in the InvFEST database (http://invfestdb.uab.cat/). In addition, inversion genotypes have been deposited in the dbVar database (https://www.ncbi.nlm.nih.gov/dbvar/) under accession number nstd169 [https://www.ncbi.nlm.nih.gov/dbvar/studies/nstd169/]. The source data underlying Figs. 2, 3a-c, 4a-c, 5a, c, e and 6a-b and Supplementary Figs. 1, 2a-b, 4a, f, 5 and 8a-c are provided as a Source Data file. All other relevant data are available upon request.

## Code availability

Computer code is available upon request.

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

## Acknowledgements

We thank Cristina Aguado, Olga Dolgova, Teresa Soos, Esteban Urrea, David Vicente-Salvador, and Roser Zaurín for help with inversion genotyping, Antonio Barbadilla, Ruth Gómez, José Ignacio Lucas-Lledó, Sebastián Ramos-Onsins, and Alfredo Ruiz for help with the evolutionary analysis, Mariona Bellet, Robert Castelo, Diego Garrido, and Roderic Guigó for help with the gene-expression analysis, Sònia Casillas and Alexander Martínez-Fundichely for help with the inversion selection, and Xavier Estivill, Tomàs Marquès-Bonet, Aurora Ruiz-Herrera, the Coriell Institute for Medical Research, the Barcelona Zoo and the Banc de Teixits Animals de Catalunya (BTAC) for providing the human and non-human primate samples used in this study. This work was supported by research grants ERC Starting Grant 243212 (INVFEST) from the European Research Council under the European Union Seventh Research Framework Programme (FP7), BFU2013-42649-P and BFU2016-77244-R funded by the Agencia Estatal de Investigación (AEI, Spain) and the European Regional Development Fund (FEDER, EU), and 2014-SGR-1346 and 2017-SGR-1379 from the Generalitat de Catalunya (Spain) to M.C., a PIF PhD fellowship from the Universitat Autònoma de Barcelona (Spain) to C.G.D., a La Caixa Doctoral fellowship to J.L.J., and a FPI PhD fellowship from the Ministerio de Economía y Competitividad (Spain) to M.O. and I.N. M.G.V. was supported in part by POCI-01-0145-FEDER-006821 funded through the Operational Programme for Competitiveness Factors (COMPETE, EU) and UID/BIA/50027/2013 from the Foundation for Science and Technology (FCT, Portugal).

## Author contributions

M.C. conceived the inversion genotyping strategy, devised the study and oversaw all the steps; S.V., M.P., and M.C. designed the genotyping assays; S.V., D.I., A.D., and M.P. carried out experiments; C.G.D., M.G.V., I.O., C.L.F., M.P., and M.C. analyzed evolutionary data; C.G.D., J.L.J., D.C., B.B., I.N., P.O., A.A., and L.F. performed selection tests; J.L.J., M.O., L.P., T.E., and M.C. analyzed functional effects; A.B. provided samples; M.C., C.G.D., J.L.J., M.G.V., L.F., and M.P. wrote the paper and all the authors contributed comments to the final version of the manuscript.

## Additional information

**Competing interests:** The authors declare no competing interests.

