## [Peer Review File · Nature Communications]

Reviewers' comments:

Reviewer #1 (Remarks to the Author):

Giner-Delgado and colleagues present a well-written and nicely designed study focused on genotyping a representative set of 45 inversions previously identified in the human genome across diverse populations. The paper describes a novel "high-throughput" genotyping approach using multiplex ligation-dependent probe amplification (MPLA) based on the principles of inverse PCR that shows relatively high concordance with previously published genotypes (n=3377) in the case of non-homologous (NH) inversions, and less so with NAHR inversions flanked by larger segmental duplications (SDs). They applied this to 1KG individuals and identified SNPs in LD with inversions through existing datasets. They find that NAHR inversions experience recurrent events so are less often tagged by flanking SNPs. Also, these same inversions also exist recurrently in nonhuman primates. For inversions with tagging SNPs, they re-analyzed and/or leveraged already existing data on SNP associations from RNA-seq eQTLs (Guevadis and GTEx) as well as GWAS studies to characterize putative "functional" impacts of genotyped inversions. Overall, to my knowledge, the study is the most comprehensive analysis to date of human inversions.

Comments:

1. The title is misleading by stating "functional" impact of inversions. Although it is true that associations exist, there is no experimental evidence that directly connects inversion alleles to a true functional outcome. True, molecular phenotypes of expression may exist, but no experimental validations were performed.
2. To distinguish this study and show the novelty of the analysis, it would be useful to move descriptions of previous inversion discovery/genotyping efforts (both methods and scale of individuals genotyped) from the Methods to the Introduction. Specifically, how were inversions identified that are now included in the InvFEST database and how many individuals genotyped previously.
3. Include more detail into why/how were the 45 polymorphic inversions chosen for genotyping (representing 50% of the <1% frequent variants) included in the Results. In the Methods it seems they were chosen based on authors' ability to design the MPLA assay. It would be useful to know, since this is a novel genotyping method, of the complete set of known inversions that exist, how many can/cannot be genotyped with the MPLA method.
4. It is unclear if any inversions associated with disease susceptibility or interesting phenotypes (e.g., 17q21.31, 15q13, etc.) were chosen for genotyping in this study. If yes, it would be useful to highlight results from these particular events. If not, why were they not included?
5. In Figure 3A, why is there discordance between fractions shared SNPs in 1KG and Hapmap?
6. The use of the Y chromosome inversion and male haplogroups is a nice/creative analysis.
7. The final Results section "Integrative analysis of genomic impact of inversions" is more suited to the Discussion. There is no new novel analysis of the data. Rather, the authors are interpreting and developing putative stories from the data.

Reviewed by Megan Dennis

Reviewer #2 (Remarks to the Author):

The authors have selected a set of 45 polymorphic inversions from the InvFEST database, with sizes ranging from 83 bp to 415 kb, and IRs mapping at the BPs of 53% of these inversions. They developed new high-throughput genotyping assays combining iPCR and MLPA, and genotyped the 45 inversions in 551 individuals from seven worldwide populations from HapMap and 1000 Genomes projects. The methods are sound and thorough with extensive documentation in the supplemental files.

Comments:

1. The manuscript shows promise but discussion and readability need to be improved. The discussion is short and not very into depth. Discussion of the results in the results paragraph would better fit into the discussion section.
2. It would be good for the authors to put this work into context of the recent manuscripts by Chaisson et al. and Sanders et al.
3. Clarify what are the limitations of the method used (size of the inversions and of the IRs...)
4. Line 45. The authors should specify what kind of genotyping method they used and the size range of the inversions that they studied.
5. Line 69. Clarify what "certain types" is (non-recurrent?)
6. Line 73. What about the Chaisson et al, Sanders et al. and Catacchio et al. manuscripts?
7. Line 102. What are the criteria for selecting those 45 inversions?
8. Lines 121-122. "Four additional inversions not initially included in the MLPA-like assays were tested directly by multiplex PCR or iPCR". Why? Does this make 49 inversions (45+4) or 41 inversions were initially selected?
9. Line 128. "3,377 published genotypes" needs a reference
10. Lines 133-134. Explain better where these 2,160 and 24,355 inversion genotypes come from.
11. Line 143. 30 CEU and 30 YRI individuals or trios? The numbers don't match... 551 total individuals tested, minus 480 unrelated individuals makes 71 related individuals.
12. Lines 932-934. The numbers don't match with the ones in the main text (481 vs 480 in the main).
13. Lines 159-161. I would remove any discussion from the Results section and add it to the Discussion.
14. Lines 208-210. What processes are the authors referring to? (recurrence?) Also here I would move any comments to the discussion section.
15. Lines 295-296. Explain why 42. Why 40 in chimpanzee and 41 in gorilla? Are there some inversions that were genotyped in one species and not in the other one and vice versa?
16. Lines 323-324. Add references.
17. Lines 379-383. Any discussion should be moved to the Discussion paragraph.
18. Line 654. The statement "which could be as small as 1 kb" needs a reference.
19. Table S1 – explain in the legend what SD means.
20. Figure S2B – on the figure is written "Genotype agreement", while in the legend is written "Genotype disagreement".
21. Figure 2, Selection panel – is not shown or described anywhere what the symbols mean
22. Table S1 – the ancestral status cannot be recurrent. I would write ND if cannot be determined because the inversion is recurrent in the species analyzed.
23. Have the paper checked by a native speaker.

Reviewer #3 (Remarks to the Author):

There is very little known about presence of large-scale inversions in the human genome. In this paper, Giner-Delgado et al present a comprehensive genotyping effort using techniques such as PCR and iMLPA on a set of 45 common polymorphic inversions on a large cohort of 551 healthy individuals. The genotyping data is then used to study selection on these variant alleles in the genome and to study the effect of the inversions on functional regions of the genome or association with phenotypic traits. Overall the paper is well written and organized.

Given that inversions are very difficult to detect and discover in genomes, we find that the approach taken by the authors to comprehensively genotype 45 known inversions in a large cohort of healthy individuals to be intriguing. This analysis is quite comprehensive and the results presented is the first ever report of experimentally generated large scale genotypes of inversions in humans. This is quite an accomplishment.

However most of the presented results are as expected from previous studies. Furthermore, recent results from several studies have discovered that a normal human genome contains a much large number of inversions (several hundred based on results out of the HGSV consortium). A large majority of these inversions are not the canonical simple inversions that are analyzed by the authors. So the overall conclusions from the paper remain quite narrow in scope in terms of overall impact of inversions to the human genome.

While very interesting, I believe that the paper remains very narrow in scope and would not recommend for publication in Nature Communications.

Response to Reviewers' comments:

We thank the reviewers for the careful assessment of the manuscript and their comments. We have done a complete revision of the text and addressed the reviewers' concerns, incorporating the necessary clarifications and additional information (within the journal length limitations). In particular, according to *Nature Communications* guidelines, we have reduced significantly the text without losing any of the information, which has contributed to improve readability and eliminate some expressions that might not be totally idiomatic. In addition, as suggested, we have extended the introduction and the discussion, and created two separate sections of Methods, which describes how all the experiments and analyses were done, and Supplementary Methods, including additional details for some specific analyses.

All the changes made from the earlier version of the manuscript can be found in the accompanying word document as track changes. Below, there is a point-by-point summary of the reviewers' comments, with our response in blue below.

Reviewer #1 comments:

Giner-Delgado and colleagues present a well-written and nicely designed study focused on genotyping a representative set of 45 inversions previously identified in the human genome across diverse populations. The paper describes a novel "high-throughput" genotyping approach using multiplex ligation-dependent probe amplification (MPLA) based on the principles of inverse PCR that shows relatively high concordance with previously published genotypes ($n=3377$) in the case of non-homologous (NH) inversions, and less so with NAHR inversions flanked by larger segmental duplications (SDs). They applied this to 1KG individuals and identified SNPs in LD with inversions through existing datasets. They find that NAHR inversions experience recurrent events so are less often tagged by flanking SNPs. Also, these same inversions also exist recurrently in nonhuman primates. For inversions with tagging SNPs, they re-analyzed and/or leveraged already existing data on SNP associations from RNA-seq eQTLs (Geuvadis and GTEx) as well as GWAS studies to characterize putative "functional" impacts of genotyped inversions. Overall, to my knowledge, the study is the most comprehensive analysis to date of human inversions.

Comments:

1. The title is misleading by stating "functional" impact of inversions. Although it is true that associations exist, there is no experimental evidence that directly connects inversion alleles to a true functional outcome. True, molecular phenotypes of expression may exist, but no experimental validations were performed.

We agree with the reviewer that, as for most genetic variants, it is often difficult to determine the causal role of the inversions in the functional effects. However, using the Geuvadis RNA-Seq data in many of the same individuals for which the inversions had been genotyped, we checked the effect on gene expression levels of 42 of the studied inversions. In some cases we were able to show that the inversion is the lead variant explaining the largest fraction of expression variation, and a few of them have a clear molecular effect in the sequence of the affected genes. In addition, although we did not directly validate these expression differences experimentally, several were replicated in different datasets generated using independent methods (such as RNA-Seq and arrays). We therefore think that, thanks to the generated genotypes and the different analyses performed, the manuscript describes the most complete study of the functional consequences that inversions might have in humans so far. In our opinion it is important to include the word "functional" in the title. However, to soften this claim we propose to rearrange the title to emphasize the evolutionary aspects and add the

word “common” to clarify that we are studying a subset of human inversions, as in “Evolutionary and functional impact of common polymorphic inversions in the human genome”.

2. To distinguish this study and show the novelty of the analysis, it would be useful to move descriptions of previous inversion discovery/genotyping efforts (both methods and scale of individuals genotyped) from the Methods to the Introduction. Specifically, how were inversions identified that are now included in the InvFEST database and how many individuals genotyped previously.

According to the reviewer suggestions, we have restructured the introduction and created two new paragraphs that describe in more detail the different studies and methods used to predict inversions in humans and add more information on previous inversion genotyping efforts, including some of the existing limitations in inversion study. Also, we have removed redundant information from the Results and Methods sections. The new paragraphs read as:

Genome-wide inversion discovery has been typically based on genome sequence comparison^{10,11} or paired-end mapping (PEM)^{4,12}, although recent studies have exploited newer techniques that could be especially useful for inversion detection, such as long-read sequencing^{13–15}, Strand-seq¹⁶, BioNano optical maps¹⁷, or a combination of them¹⁸. In most cases around 100-200 inversions have been predicted, with a maximum of 786 predictions in the 1000 Genomes Project (1000GP)^{4,19}. However, these methods are not suitable for high-throughput genotyping, and with few exceptions^{4,16,20}, just a reduced number of individuals (1-15) have been analyzed. Moreover, the presence of repetitive sequences at the breakpoints influences the inversions that can be detected by each technique and results in high error rates for inversion validation compared to other variants^{4,18,20,21}.

Apart from the intensely-studied 17q21.31 and 8p23.1 inversions^{22,23}, genotyping efforts have been restricted to a small number of inversions and samples. For example, four other large inversions have been genotyped by FISH²⁴ and five smaller inversions by PCR²⁵ in 27 and 42 individuals of four populations, respectively. In addition, PCR and inverse PCR (iPCR) have been used for targeted studies of 34 inversions in 70-90 Europeans^{21,26} and a more worldwide characterization of three inversions^{25,27,28}. Also, although inversion genotypes might be predicted based on SNP data, these methods can only detect inversions above a certain size or associated with specific SNP combinations and the error rate can be high^{21,23,25,29}. Therefore, it is not yet clear how many polymorphic inversions really exist in humans and very little is known about their global frequency and distribution¹⁹.

3. Include more detail into why/how were the 45 polymorphic inversions chosen for genotyping (representing 50% of the <1% frequent variants) included in the Results. In the Methods it seems they were chosen based on authors' ability to design the MPLA assay. It would be useful to know, since this is a novel genotyping method, of the complete set of known inversions that exist, how many can/cannot be genotyped with the MPLA method.

As suggested, we have moved some of the details of the inversion selection from the Methods to the Results. Basically, we tried to include as many inversions as possible from those that were predicted at the time and that could be experimentally validated as real inversions, most of which have been later detected in other studies. Although the total number of inversions and their frequencies in human populations is still unclear, we had previously used the 1000GP dataset as a reference and have now updated the information with the recent analysis from Audano et al. (2019) in 15 individuals. This allowed us to estimate that the inversions in our study represent approximately half of those with >5% frequency in human populations, and a specific mention to the common nature of these inversions has been included in the title as well. To illustrate that these inversions are a representative sample of

those predicted, we have added the information of which of the 45 studied inversions were predicted by different studies in Supplementary Data 1. Finally, we have included information on the limitations for inversion genotyping. The paragraph now reads:

We focused on a representative set of 45 paracentric inversions from the InvFEST database¹⁹, which comprised most of those experimentally validated by PCR-based techniques when the project started¹⁹ and corresponds approximately to half of the estimated number of real variants with >5% frequency in human populations^{4,14} (Supplementary Data 1; Supplementary Fig. 1). These inversions were originally detected in the comparison of the hg18 and HuRef genome assemblies¹⁰ or a fosmid PEM survey in nine individuals¹², and between 1 and 36 of them have also been identified in different recent studies (Supplementary Data 1). The main limitation for inversion genotyping was due to breakpoint IRs, that had to be of less than 25-30 kb and with target sites for certain restriction enzymes at both sides but not within the IRs²⁶, which excluded previously-known large inversions mediated by big repeat blocks¹⁹.

On the other hand, in the discussion we have added several sentences talking about the problems of inversion genotyping and giving an estimate of how many other inversions from InvFEST could be interrogated. We now also discuss possible additional technical improvements to complete the picture of human inversions.

Despite the effort to include as many inversions as possible, only those validated and with well-characterized breakpoints could be analyzed¹⁹. In addition, important limitations exist for the study of those mediated by IRs. Although it is not clear how many of them are real, in the InvFEST database there are ~100 inversion predictions with 1-25 kb inverted segmental duplications (SDs) at the breakpoints, which could be potentially interrogated using our method, and ~250 with larger SDs, which are currently out of reach of the methodology¹⁹. Therefore, further improvements and complementary strategies, such as the possibility of making directed cuts in specific positions or estimating the distance between regions separated by larger repeats, are necessary to expand the range of analyzed inversions.

4. It is unclear if any inversions associated with disease susceptibility or interesting phenotypes (e.g., 17q21.31, 15q13, etc.) were chosen for genotyping in this study. If yes, it would be useful to highlight results from these particular events. If not, why were they not included?

Unfortunately these inversions are characterized by having extremely large blocks of SDs at the breakpoints and it is not possible to genotype them experimentally by any technique other than FISH. The answer to this comment is included in the paragraph about the genotyping technique limitations mentioned in our response to Comment 3 above:

The main limitation for inversion genotyping was due to breakpoint IRs, that had to be of less than 25-30 kb and with target sites for certain restriction enzymes at both sides but not within the IRs²⁶, which excluded previously-known large inversions mediated by big repeat blocks¹⁹.

5. In Figure 3A, why is there discordance between fractions shared SNPs in 1KG and Hapmap?

The difference in the fraction of shared SNPs from HapMap and 1000GP data derives from the different density and frequency distribution of the SNPs in each study. 1000GP gives a more complete view of nucleotide variation, and contains a higher number of SNPs, including many at low frequency found in a few individuals, although the low sequencing coverage can lead to errors. HapMap

genotypes might be more accurate but correspond to just a few relatively-frequent SNPs, which tends to inflate the proportion of shared SNPs for many inversions. The total number of SNPs analyzed in each case can be found in Supplementary Table 2, but to clarify this point we have added an explanation in Figure 3 legend:

...fraction of shared SNPs inside the inverted region from the total number of SNPs analyzed in 1000GP or HapMap (which includes less SNPs and results in higher shared fractions),...

6. The use of the Y chromosome inversion and male haplogroups is a nice/creative analysis.

We thank the reviewer for her positive comments.

7. The final Results section “Integrative analysis of genomic impact of inversions” is more suited to the Discussion. There is no new novel analysis of the data. Rather, the authors are interpreting and developing putative stories from the data.

As recommended by both Reviewers 1 and 2, we have moved this part to the Discussion and have shortened it considerably to adhere to the journal length limitations.

We have found 23 and 22 inversions with different selection and functional signatures, respectively (Fig. 2). More importantly, although not all analyses could be applied to every inversion, there is a significant enrichment of inversions with both effects on genes or gene expression and selection signals directly linked to them (Fisher’s exact test $P = 0.0320$) (see Supplementary Methods). This combination of the two independent types of evidence strongly indicates that inversions can have important consequences in humans.

One particularly interesting example is HsInv0201, an old inversion (>1.5 Mya) with intermediate frequency around the globe and clear signals of balancing selection (Fig. 6A), which deletes an exon of SPINK14 and is lead eQTL for two nearby genes (Supplementary Data 9). Moreover, the inversion haplotype is the main responsible for the lower SPINK6 expression during immune response to Salmonella infection^{52,53} (Fig. 6B). In fact, HsInv0201 eliminates the promoter and first exon of a putative novel SPINK6 isoform (Fig. 6B) and it is in high LD ($r^2 = 0.971$ in EUR) with SNPs accounting for plasma levels of SPINK6 protein⁵⁴. Together with the role of several of the affected genes in lung and extracellular mucosae, this suggests that the inversion could be related to immune response. In HsInv0006, its particular distribution pattern and the selection test results point to positive selection of the derived allele in Africa (Fig. 6A). Furthermore, the inversion is located within DSTYK first intron²¹ and is associated with expression changes in the proximal genes, including DSTYK upregulation in different tissues (Fig. 6C; Supplementary Fig. 5). DSTYK deletion causes pigmentation problems and elevated cell death after ultraviolet irradiation⁵⁵. Thus, positive selection on these traits could explain the inversion increase in Africa. Incidentally, the inverted orientation has been linked to higher risk of glaucoma in Europeans (Supplementary Table 4) and glaucoma is more common and severe in individuals from African ancestry⁵⁶. Other interesting candidates include HsInv0031, HsInv0059, HsInv0124, HsInv0340 and HsInv0389 (Fig. 2).

Reviewed by Megan Dennis

Reviewer #2 comments:

The authors have selected a set of 45 polymorphic inversions from the InvFEST database, with sizes ranging from 83 bp to 415 kb, and IRs mapping at the BPs of 53% of these inversions. They

developed new high-throughput genotyping assays combining iPCR and MLPA, and genotyped the 45 inversions in 551 individuals from seven worldwide populations from HapMap and 1000 Genomes projects. The methods are sound and thorough with extensive documentation in the supplemental files.

Comments:

1. The manuscript shows promise but discussion and readability need to be improved. The discussion is short and not very into depth. Discussion of the results in the results paragraph would better fit into the discussion section.

As mentioned above, we have restructured the text and reduced it considerably to adhere to the journal length limitations and to improve readability. In addition, we have extended the Discussion section by including some of the information placed previously in the Results and adding a new paragraph about inversion genotyping and its limitations (see response to Reviewer 1 Comment 3). Finally, as suggested by Reviewers 1 and 2 we have incorporated into the Discussion the integrative summary of different promising results for specific inversions (see response to Reviewer 1 Comment 7).

2. It would be good for the authors to put this work into context of the recent manuscripts by Chaisson et al. and Sanders et al.

As suggested by the reviewers, we have broadened the introduction to include more detail on previous work on inversion prediction and genotyping. We already cited the Sanders et al. (2016) article, and we have included additional references to the work of Chaisson et al. (2019) and Audano et al. (2019), both of which have been published in the last months. Specifically, we emphasize that each article predicts around 100-200 inversions, but in all cases the number of genotyped individuals is small, so there is the need to generate accurate inversion genotyping data in higher numbers of individuals to be able to determine their functional and evolutionary impact. See response to Reviewer 1 Comment 2 for new Introduction paragraph.

3. Clarify what are the limitations of the method used (size of the inversions and of the IRs...)

As mentioned in the response to Reviewer 1 Comment 2 and 3, we now describe and discuss the limitations of the method used for inversion genotyping in the Results and the Discussion:

Results: The main limitation for inversion genotyping was due to breakpoint IRs, that had to be of less than 25-30 kb and with target sites for certain restriction enzymes at both sides but not within the IRs²⁶, which excluded previously-known large inversions mediated by big repeat blocks¹⁹.

Discussion: In addition, important limitations exist for the study of those mediated by IRs. Although it is not clear how many of them are real, in the InvFEST database there are ~100 inversion predictions with 1-25 kb inverted segmental duplications (SDs) at the breakpoints, which could be potentially interrogated using our method, and ~250 with larger SDs, which are currently out of reach of the methodology¹⁹. Therefore, further improvements and complementary strategies, such as the possibility of making directed cuts in specific positions or estimating the distance between regions separated by larger repeats, are necessary to expand the range of analyzed inversions.

4. Line 45. The authors should specify what kind of genotyping method they used and the size range of the inversions that they studied.

We have added the suggested information:

Here, we develop a new high-throughput genotyping method based on probe hybridization and amplification, and we perform a complete study of 45 common human inversions of 0.1-415 kb.

5. Line 69. Clarify what “certain types” is (non-recurrent?)

We now clarify that inversion prediction based on SNP data requires inversions of enough size and linked to particular SNP combinations:

Also, although inversion genotypes might be predicted based on SNP data, these methods can only detect inversions above a certain size or associated with specific SNP combinations and the error rate can be high^{21,23,25,29}.

6. Line 73. What about the Chaisson et al, Sanders et al. and Catacchio et al. manuscripts?

In this particular sentence we were considering studies in which inversions were genotyped in multiple individuals. Now with the reorganization of the Introduction we have created two separate paragraphs. The first one deals with the use of different techniques to predict inversions and contains references to all these articles. The second paragraph describes the available data on inversion genotypes that have been independently validated. See response to Reviewer 1 Comment 2 for new Introduction paragraphs.

7. Line 102. What are the criteria for selecting those 45 inversions?

As explained in more detail in the response to Reviewer 1 Comment 3, we have moved the information on the inversion selection from Methods to Results. Basically, we included all the inversions that could be experimentally validated at the time and that were mainly predicted in two studies, the comparison of the Hg18 and HuRef genomes and Kidd et al. (2008) study of paired-end mapping in 9 individuals.

8. Lines 121-122. “Four additional inversions not initially included in the MLPA-like assays were tested directly by multiplex PCR or iPCR”. Why? Does this make 49 inversions (45+4) or 41 inversions were initially selected?

We actually selected a total of 45 inversions: 41 were genotyped with the initial MLPA assays, while 4 more that could not be genotyped by MLPA due to problems with the probe design were later genotyped directly by PCR in an effort to make the study as comprehensive as possible. The genotyping strategy is illustrated in Fig. 1 and we have tried to clarify this point more by moving upwards the number of inversions genotyped by each method within the figure and by specifying in the text that 41 of the selected inversions were genotyped by MLPA:

Of those, 41 inversions were genotyped simultaneously using high-throughput assays derived from the multiplex ligation-dependent probe amplification (MLPA) technique⁴².

9. Line 128. “3,377 published genotypes” needs a reference

Due to space constraints, we have included the information of the number of individuals genotyped by different previous studies in a new column within Supplementary Data 1 and explicitly refer the reader to the information available in this file and the InvFEST database:

MLPA and iMLPA inversion genotypes were carefully validated through several analyses and quality controls (Fig. 1): (1) comparison with 3,377 available genotypes¹⁹ (see Supplementary Data 1 for data source);...

10. Lines 133-134. Explain better where these 2,160 and 24,355 inversion genotypes come from.

We have removed the reference to the total number of 24,355 inversion genotypes generated, which is already included in Fig. 1. Also, we explain in more detail that 2,160 additional genotypes were validated in this work by PCR or iPCR, which as before are listed in the new column within Supplementary Data 1 with the number of validated genotypes for each inversion:

As part of the validation, 2,160 extra genotypes with discrepancies or possible errors, plus other randomly selected genotypes, including the whole set of 551 individuals for the three inversions that accumulated the highest error rate (HsInv0045, HsInv0055 and HsInv0340), were tested by PCR or iPCR (Supplementary Data 1).

11. Line 143. 30 CEU and 30 YRI individuals or trios? The numbers don't match... 551 total individuals tested, minus 480 unrelated individuals makes 71 related individuals.

We apologize for the confusion. Actually the 30 CEU and 30 YRI refer to trios (total of 90 individuals in each case) and there were 10 additional related individuals with cryptic relationships that were later identified by the 1000GP. In addition, since the population of origin of individual NA15510 is not known, it has not been included in the calculation of the population frequencies. We have tried to explain this more clearly in the new Methods section:

We used 550 human samples included in the last phase of the HapMap Project and many of them in 1000GP phase 1 (Ph1) and 3 (Ph3)¹⁻³, which belong to seven populations of four main population groups: Africa (AFR) (YRI and LWK), Europe (EUR) (CEU and TSI), South-Asia (SAS) (GIH) and East Asia (EAS) (CHB and JPT), plus individual NA15510 of unknown origin (see Supplementary Table 1 and Supplementary Data 2 for details). Most individuals were unrelated, but 70 are either children of mother-father-child trios (30 in YRI and 30 in CEU) or cryptic first and second-degree relatives (9 in LWK and 1 in GIH)^{2,3}.

12. Lines 932-934. The numbers don't match with the ones in the main text (481 vs 480 in the main).

We apologize again for the confusion, which as before was due to the inclusion of individual NA15510 of unknown origin. As can be seen in the new Methods paragraph above, we now do not state the number of 481 unrelated individuals to avoid confusion.

13. Lines 159-161. I would remove any discussion from the Results section and add it to the Discussion.

As suggested, we now just mention this in the Discussion.

14. Lines 208-210. What processes are the authors referring to? (recurrence?) Also here I would move any comments to the discussion section.

Yes, we were referring to recurrence, but as suggested we now just mention this in the Discussion.

15. Lines 295-296. Explain why 42. Why 40 in chimpanzee and 41 in gorilla? Are there some inversions that were genotyped in one species and not in the other one and vice versa?

Yes, the problem was that the genotyping assays for a few inversions just worked in one species or the other or simply that some inversion regions were deleted in one of the species. This is now explained in more detail in the Results and new Methods section, with the information of the inversions genotyped in each species listed in Supplementary Data 5:

Results: The published data on the ancestral orientation of 32 of the inversion regions^{21,26,28} was complemented and expanded by experimentally testing 42 inversions for which the human or modified assays generated reliable results in a panel of 23 chimpanzees (40 genotyped inversions) and 7 gorillas (41 genotyped inversions) (Supplementary Data 5).

Methods: In some cases, new chimpanzee or gorilla specific primers and restriction enzymes for iPCR were used to overcome the human assay problems²⁶ (Supplementary Data 5; Supplementary Data 11). However, this was not always possible due mainly to deletions or genome gaps, and a few inversions in one or both species could not be tested.

16. Lines 323-324. Add references.

Due to space constraints, we have moved the mention to the specific genomes analyzed to Supplementary Data 6, where we have included the references to the publications describing each of the genome sequences. The sentence now reads:

Moreover, we checked the presence of the breakpoint sequences of 19 NH inversions without IRs in available Neanderthal, Denisovan and two ancient modern human genomes (Supplementary Data 6).

17. Lines 379-383. Any discussion should be moved to the Discussion paragraph.

We have reduced this part significantly and as suggested we have removed the last sentences from the Results and just included the main points in the Discussion.

18. Line 654. The statement "which could be as small as 1 kb" needs a reference.

Actually the statement was a preliminary estimation of the possible selective effects against inversions based on the negative fertility costs associated to the generation of unbalanced gametes by recombination in heterozygote individuals. The calculation of this cost depends on the recombination rate in each region and how negative the generation of aberrant gametes is for the fitness of the individual. Therefore, it requires a more formal description that it is out of the scope of this paper. With the reorganization of the discussion we have eliminated the reference to a specific value and we just summarize the general idea that as supported by the negative correlation between inversion genetic length and frequency longer inversions are expected to have negative effects:

Inversions differ from other genetic variants because of their expected negative consequences in fertility resulting from the generation of unbalanced gametes by recombination^{30,31}, which is exemplified by the reduction in frequency with genetic length (Fig. 4B) and the small number of inversions described compared to CNVs¹⁹. According to this, there could be a maximum length for an inversion to behave neutrally in terms of its fertility effects. Above that size, some type of compensatory selection, perhaps related to advantageous regulatory changes on nearby genes, will be necessary for the inversions to reach a certain frequency.

19. Table S1 – explain in the legend what SD means.

We have updated the legend of Supplementary Data 1 as:

...using whenever possible the annotations for segmental duplications (SDs) and transposable elements (TEs) from the UCSC genome browser

20. Figure S2B – on the figure is written “Genotype agreement”, while in the legend is written “Genotype disagreement”.

We apologize for the inconsistency. The legend now reads:

b. Genotype agreement between the 14 inversions in common with the 1000GP structural variant release¹ according to the InvFEST database² for the 434 samples shared in both datasets.

21. Figure 2, Selection panel – is not shown or described anywhere what the symbols mean.

We apologize for the missing information. The legend now states:

Populations where the signal was detected are indicated by different colors in the corners of each cell, with alternating + and × symbols to avoid visual overlap.

22. Table S1 – the ancestral status cannot be recurrent. I would write ND if cannot be determined because the inversion is recurrent in the species analyzed.

As suggested by the reviewer, both in updated Supplementary Data 1 and Supplementary Data 6 we have labeled these inversions as ND, indicating in parentheses that they are recurrent in other primates, to distinguish them from other cases in which no results were obtained.

23. Have the paper checked by a native speaker.

During the reduction of the manuscript, different parts of the text have been rewritten and some of the less natural or idiomatic expressions have been removed. Also, the manuscript has been extensively revised by several of the authors and most readability problems should now be solved.

Reviewer #3 comments:

There is very little known about presence of large-scale inversions in the human genome. In this paper, Giner-Delgado et al present a comprehensive genotyping effort using techniques such as PCR and iMLPA on a set of 45 common polymorphic inversions on a large cohort of 551 healthy individuals. The genotyping data is then used to study selection on these variant alleles in the genome

and to study the effect of the inversions on functional regions of the genome or association with phenotypic traits. Overall the paper is well written and organized.

Given that inversions are very difficult to detect and discover in genomes, we find that the approach taken by the authors to comprehensively genotype 45 known inversions in a large cohort of healthy individuals to be intriguing. This analysis is quite comprehensive and the results presented is the first ever report of experimentally generated large scale genotypes of inversions in humans. This is quite an accomplishment.

However most of the presented results are as expected from previous studies. Furthermore, recent results from several studies have discovered that a normal human genome contains a much large number of inversions (several hundred based on results out of the HGSV consortium). A large majority of these inversions are not the canonical simple inversions that are analyzed by the authors. So the overall conclusions from the paper remain quite narrow in scope in terms of overall impact of inversions to the human genome.

While very interesting, I believe that the paper remains very narrow in scope and would not recommend for publication in Nature Communications.

We appreciate the interest in the work and the reviewer positive comments, highlighting the unique nature of the work and the accomplishment it represents. We also agree that ideally the number of inversions analyzed should be higher, but as the reviewer mentions there is very little experimentally-generated data on human inversion genotypes, which is precisely due to the complication of their study and we have now tried to clarify this more in the introduction. As an example, a recent study to characterize all types of structural variants in 15 genomes by Audano et al (2019) was able to genotype many of the variants in multiple additional individuals, but no genotypes were generated for the ~200 predicted inversions. Similarly, the article just published by Chaisson et al. (2019) describing the work of the HGSV Consortium to characterize structural variants in 9 individuals using different techniques to maximize sensitivity predicts 308 inversions, of which 227-229 are classified by the authors as simple and the others are apparently inverted duplications, which cannot be considered really as inversions (or at least they are not expected to have the same properties as the other ones). In this case no additional genotypes are generated for the inversions either. In addition, as an example of the high error rates in inversion prediction, both studies include a certain amount of inversions classified as false in the InvFEST database, such as genome assembly errors, and it is still necessary to check some of the more complex events in detail. Therefore, the total number of predicted inversions is actually similar to that of other studies. In fact, the 45 inversions analyzed here comprise a significant fraction of those predicted by Audano et al (2019) and Chaisson et al. (2019) (especially considering the common ones), and 75-80% of them are included in each dataset, which emphasizes that they are representative of the inversions present in the human genome and the relevance of the data we have generated. It is also important to mention that we have analyzed different types of inversions, including not just those with simple breakpoints, but many flanked by other indels or inverted repeats (IRs).

On the other hand, thanks to the genotypes generated, we have carried out an exhaustive analysis of the largest experimentally-validated dataset on human inversions, which had not been possible before. The only comparable analysis in multiple individuals is that of the 1000GP, that despite the impressive effort, missed two thirds of the inversions studied here, including most of those mediated by IRs, and due to the low-coverage failed to genotype accurately many of the inversions, which complicates obtaining reliable population-level conclusions.. In our case, we describe the development

of a new method to genotype inversions that can be used in future studies of the association of these variants with phenotypic traits and disease susceptibility in humans. In addition, our method focuses especially in inversions mediated by IRs, which are those most difficult to study by other techniques and the ones whose effects might have been missed due to the low linkage disequilibrium with SNPs. Specifically,, we have been able to quantify in detail a great degree of recurrence for inversions at different levels, that, although it has been described for some inversions in smaller datasets before, we do not think it was expected. In addition, the complete analysis of the selective forces acting on inversions has allowed us to find different candidate inversions to being selected during human evolution. Finally, we have also carried out a complete functional analysis combining different types of information for the first time in most of the studied inversions. This analysis suggests that several inversions could have important consequences both in gene expression and phenotypic traits. Moreover, we have observed a significant enrichment of inversions with evidence of the same time of selection and functional effects, which together strongly support their role in genome evolution and function. Therefore, we do think that this study is an important contribution to the understanding of this difficult to characterize type of variation in humans.

REVIEWERS' COMMENTS:

Reviewer #1 (Remarks to the Author):

Authors have appropriately addressed all of my comments/concerns.

Reviewer #2 (Remarks to the Author):

I am grateful to the authors for their revised version of this manuscript, and their responses to all reviewers, which together provided greater clarity and provided answers to all my questions. I have no further comments.